# KARE-RAG: Knowledge-Aware Refinement and Enhancement for RAG

## Abstract

Retrieval-Augmented Generation (RAG) enables Large Language Models (LLMs) to access external knowledge sources, significantly enhancing their ability to perform knowledge-intensive tasks. As RAG systems become increasingly vital for real-world applications, improving their ablility of leveraginge externel knowledge has emerged as a critical research direction. Recent studies have explored fine-tuning approaches to enhance LLMs' adaptability in RAG scenarios. However, the inherent complexity of RAG systems makes it challenging to achieve robust performance without substantial amounts of high-quality training data, particularly when handling multi-hop reasoning and other complex tasks that require rich information content and intricate relationships between knowledge elements.

In this paper, we present KARE-RAG (Knowledge-Aware Refinement and Enhancement for RAG), which introduces a novel strategy for RAG optimization. Our key insight is that by training models to construct structured knowledge representations as intermediate outputs, models can learn robust information discrimination strategies from minimal training data while maintaining the flexibility of end-to-end generation. Experiments show our method achieves superior OOD performance while requiring substantially less training data. Notably, these improvements are achieved without compromising general capabilities or requiring modifications to standard RAG inference pipelines. Our findings establish that targeted training strategies focusing on knowledge organization can unlock more efficient optimization pathways for RAG systems. All data and code will be publicly available on Github.

## 1 Introduction

Retrieval-Augmented Generation (RAG) Lewis et al. (2020); Shi et al. (2023) has become a cornerstone for knowledge-intensive tasks, enabling large language models (LLMs) to effectively combine parametric knowledge with real-time information retrieval. Despite its widespread adoption, RAG systems face inherent challenges in processing retrieved documents. The evidence is often fragmented across multiple sources, requiring models to piece together complete information from partial segments Wu et al. (2024); Xu et al. (2024a). Additionally, retrieved documents may present irrelevant or conflicting claims that need careful reconciliation Gao et al. (2023); Yoran et al. (2023); Xu et al. (2024b); Longpre et al. (2022); Liu et al. (2024a), even state-of-the-art retrieval systems cannot guarantee perfect relevance in returned documents.

Existing research has made significant progress through improving retrieval quality Jiang et al. (2023); Gao et al. (2023); Trivedi et al. (2022a); Mao et al. (2024); Ma et al. (2023); Jeong et al. (2024) or developing post-processing verification mechanisms Yu et al. (2023); Xu et al. (2023); Fang et al. (2024). Recent studies demonstrate that optimizing the generation module's ability to process and integrate multi-source knowledge represents an equally important and complementary approach Singh et al. (2021); Lin et al. (2023); Asai et al. (2023); Li et al. (2024a).

To optimize the generation model, research has evolved from supervised fine-tuning (SFT) approaches using final answers Lin et al. (2023); Asai et al. (2023) to more sophisticated preference-based methods like Direct Preference Optimization (DPO) Rafailov et al. (2024). While SFT provides a straightforward optimization baseline, its limitations in handling complex RAG scenarios—particularly the tendency toward overfitting and catastrophic forgetting—have driven the

adoption of DPO. The demonstrated effectiveness of DPO in aligning language models Ouyang et al. (2022) has established its superiority, with recent studies specifically validating its advantages in RAG scenarios through stable pairwise preference learning Li et al. (2024a).

However, applying DPO to RAG systems faces several key challenges. The end-to-end optimization paradigm provides insufficient supervision for intermediate knowledge processing steps, where critical errors in information filtering and integration often occur. This is particularly problematic in complex scenarios requiring multi-hop reasoning or conflict resolution. Additionally, when dealing with intricate queries or retrieved documents, conventional sampling methods often fail to produce effective positive examples. These limitations often necessitate impractically large training datasets to achieve robust performance Li et al. (2024a), highlighting the need for more targeted approaches to guide knowledge processing in RAG systems.

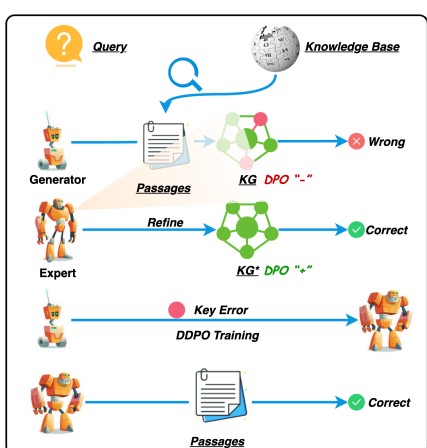

Figure 1: A Schematic Flowchart of the KARE-RAG Optimization Method.

To address these issues, we develop KARE-RAG (Knowledge-Aware Refinement and Enhancement for RAG) — an innovative training strategy that hat teaches models to better utilize retrieval documents, the schematic flowchart of our method can be viewed in Figure 1. At the core of our approach is a carefully designed knowledge representation structure that enables models to explicitly organize and express their understanding of retrieved information, enabling more precise localization of erroneous information points. To effectively generate positive training sample, we develop an automated refinement pipeline. This pipeline leverages advanced language models to correct errors while preserving the semantic and structural integrity of the original representation, thereby generating high-quality training examples that differ only in critical error regions. To better optimize the model, we employ Dense Direct Preference Optimization (DDPO)Yu et al. (2024), which enhances standard DPO by concentrating the training signal on the most important tokens - those corresponding to factual discrepancies in knowledge processing. The resulting framework teaches models to distinguish subtle but crucial differences between accurate and flawed knowledge representations, leading to more robust performance in handling retrieval contexts.

Through comprehensive empirical evaluation, our approach demonstrates statistically improvements across both in-domain and out-of-domain benchmarks, evidencing enhanced robustness in knowledge utilization and superior generalization capabilities. Notably, these improvements are achieved using standard Vanilla RAG pipelines during inference, requiring no additional computational overhead or architectural modifications to existing systems. This practical advantage ensures seamless integration with current RAG implementations while delivering measurable performance gains. Furthermore, we systematically evaluate multiple knowledge representation formats—including knowledge graph, keypoint structure, and summary—revealing that structured representations yield substantially better performance compared to unstructured summarization baselines. Crucially, the method maintains efficacy across varying model scales, achieving consistent performance gains without compromising baseline capabilities on general language understanding tasks.

## 2 RELATED WORK

LLMs possess a robust in-context learning capabilityDong et al. (2022); Ram et al. (2023), enabling them to handle more complex tasks. Retrieval-Augmented Generation (RAG) enhances LLMs by incorporating external knowledge retrievalKarpukhin et al. (2020); Xiong et al. (2020), improving performance in tasks like question answeringTrivedi et al. (2022a) and fact verification Lewis et al. (2020). It can reduce the model's propensity for hallucination and increase the reliability of the model's outputsKandpal et al. (2023); Xu et al. (2023); Luo et al. (2023). However, retrieval often

brings noisy or conflicting information, causing factual inaccuraciesLongpre et al. (2021); Xu et al. (2024a); Liu et al. (2024a).

To mitigate the impact of noise on generation effectiveness, some studies have introduced additional steps to enhance retrieval precision. For instance, several methods employ query rewrite techniques to increase the similarity between the query and the relevant textsWang et al. (2023); Mao et al. (2023), some others prompt LLMs to summarize query related informations into noteYu et al. (2023).

Optimizing the RAG process to enhance generation outcomes is also a highly popular direction. Many current efforts are exploring various methods to construct training data for training retrieval models to improve retrieval accuracyLin et al. (2023); Shi et al. (2023). Self-RAG trains models to learn to retrieve relevant documents on demandAsai et al. (2023). DDR constructs a multi-level training framework based on the final generation outcomes, simultaneously training both retrieval and generation models, enabling LLMs to learn to handle conflicts between intrinsic knowledge and retrieved documentsLi et al. (2024a).

Reinforcement learning is a commonly used algorithm in the optimization of large models. Due to the complexity of the PPO algorithmSchulman et al. (2017), current work predominantly relies on the Direct Preference Optimization(DPO) algorithmRafailov et al. (2024), which aligns the outputs of LLMs with human preferences to optimize large models. There are also numerous efforts to further refine the DPO algorithm to enhance training effectiveness. RLHF-VYu et al. (2024) increases the weight of tokens in the differing parts of positive and negative examples during training, improving the training outcomes for visual models. Step-DPOLai et al. (2024) focuses on step-by-step optimization for long-chain reasoning processes. Additionally, some studies have explored the relationship between DPO and SFT LossLiu et al. (2024b), demonstrating that SFT Loss can serve as a regularization term to stabilize the training effects of DPO.

Knowledge Graphs (KG)Hogan et al. (2021) are widely used in NLP tasks such as question answering, reasoningChen et al. (2020), and fact verification Tong et al. (2019). A significant advantage of knowledge graphs is their structured nature, which can effectively reduce noise in natural language texts, benefiting the RAG scenario. Consequently, numerous RAG works based on knowledge graphs have emergedPeng et al. (2024); Edge et al. (2024); Hu et al. (2024); Mavromatis & Karypis (2024). Current research on knowledge graphs primarily focuses on utilizing them to encode text libraries to enhance retrieval effectivenessLi et al. (2023); Huang et al. (2023); Li et al. (2024b), or integrating GNN with LLMs in the generation phase to improve output qualityGao et al. (2024); He et al. (2024); Jiang et al. (2022). These methods typically require preliminary processes to construct static knowledge graphs, making the overall workflow relatively complex.

## 3 METHOD

In this section, we introduce the Knowledge-Aware Refinement and Enhancement for RAG (KARE-RAG) method, shown in Figure 2. The core of our method is Knowledge-Aware Sampling that decomposes the generation process to explicitly model how retrieved documents are organized and utilized, transforming them into structured knowledge representations for enhanced interpretability and control. To ensure high-quality training data, we develop an advanced LLM-powered refinement pipeline that automatically corrects errors in negative examples while preserving their structural consistency with positive counterparts. For model optimization, we adopt Dense Direct Preference Optimization (DDPO)Yu et al. (2024), which extends standard DPO through token-level weighting mechanisms. This method enables the model to concentrate more effectively on discriminative features distinguishing positive and negative samples. The following sections detail each component.

### 3.1 KNOWLEDGE-AWARE SAMPLING

In a standard Retrieval-Augmented Generation (RAG) process, the LLM is given a query $q$ and a set of retrieved documents $D = \{d_1, \ldots, d_n\}$ relevant to $q$ to generate answer to the given query. Retrieved documents often contain noisy information that adversely affects generation quality.

To understand how the model processes and utilizes retrieved documents, we introduce an intermediate Knowledge-Aware Sampling step (left part of Figure 2). Before generating final answers,

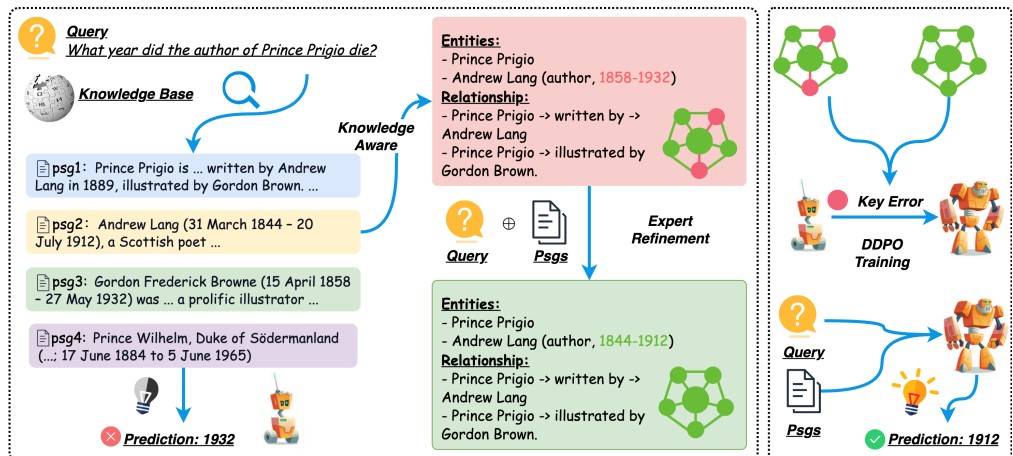

Figure 2: Illustration of our KARE-RAG optimization method.

we prompt the model to organize relevant content into structured knowledge graphs, explicitly revealing its interpretation of the source materials.

$$y_{\text{KG}} = \text{LLM}(\text{Instruct}_{\text{KG}}, q \oplus D) \tag{1}$$

where $\oplus$ denote the concatenation operation, $\text{Instruct}_{\text{KG}}$ is prompt specifically designed for Knowledge-Aware representation step.

During the Knowledge-Aware Sampling, we formalize the extracted information as structured graph representations inspired by knowledge graph architectures. This graph-based formalism provides several representational advantages, including explicit node-edge relationships that enforce logical consistency, discrete knowledge units enabling precise error localization, and topological constraints that prevent information entanglement. The resulting structured representation serves as an intermediate knowledge substrate that combines human-interpretability with machine-tractability, thereby establishing a robust foundation for subsequent processing stages while maintaining the semantic integrity of the original content.

## 3.2 REFINEMENT PIPELINE

Large language models typically produce structurally sound outputs, but may contain localized factual inaccuracies when processing retrieved documents. For example, as shown in Figure 2, while the model correctly identifies the main entities, and their relationships, it might make specific errors like incorrectly stating Andrew Lang's year of death. A fundamental challenge arises when sampling both positive examples using the same base model ($\text{LLM}_{\text{Gen}}$), as the model's stochastic generation process introduces unintended variations despite its theoretical capacity to produce corrected outputs. These variations manifest primarily in knowledge sequencing patterns and surface-level phrasing, with the divergence becoming particularly pronounced in lengthy outputs. Consequently, the perceived differences between positive ($y_{\text{KG}}^{+}$) and negative ($y_{\text{KG}}^{-}$) pairs become artificially inflated beyond their substantive errors, complicating the learning signal during optimization.

Our framework addresses this through targeted refinement. By employing an advanced model ($\text{LLM}_{\text{Exp}}$) to systematically edit $y_{\text{KG}}^{-}$ while preserving its structural backbone, we ensure contrastive pairs differ only in critical error regions. This design guarantees minimal semantic divergence—essential for the model to learn error correction patterns rather than superficial variations. As detailed in Algorithm 1, employs a multi-stage verification process to ensure structural fidelity in knowledge organization tasks. Beginning with error detection through answer validation ($y_{\text{err}} \neq y_{\text{gnd}}$), the pipeline first assesses document adequacy before proceeding to knowledge graph revision. The core refinement mechanism involves iterative context-aware patching through $\text{LLM}_{\text{Exp}}$, where the original flawed representation ($y_{\text{KG}}^{-}$) is progressively corrected using both the ground truth ($y_{\text{gnd}}$) and retrieved documents ($D$) as reference. This process continues until either successful generation of a valid final answer ($y_{\text{Gen}} = y_{\text{gnd}}$) or reaching the maximum iteration limit.

---

**Algorithm 1** Refinement Framework

---

1: Generate initial output $y_{\mathrm{KG}}^-$ using $\mathrm{LLM}_{\mathrm{Gen}}$ with Knowledge-Aware Sampling
2: **if** $y_{\mathrm{KG}}^-$ produces incorrect answer $y_{\mathrm{err}} \neq y_{\mathrm{gnd}}$ **then**
3:     Check document adequacy: $\mathrm{LLM}_{\mathrm{Exp}}(q \oplus D \oplus y_{\mathrm{gnd}})$
4:     **if** Document $D$ is sufficient **then**
5:         Revise knowledge organization with an advanced model $\mathrm{LLM}_{\mathrm{Exp}}$:

$$y_{\mathrm{KG}}^+ \leftarrow \mathrm{LLM}_{\mathrm{Exp}}(\mathrm{Instruct}_{\mathrm{revise}}, q \oplus D \oplus y_{\mathrm{gnd}} \oplus y_{\mathrm{KG}}^-)$$

6:         Generate final answer $y_{\mathrm{Gen}}$ using $y_{\mathrm{KG}}^+$ with $\mathrm{LLM}_{\mathrm{Gen}}$
7:         **while** $y_{\mathrm{Gen}} \neq y_{\mathrm{gnd}}$ & *iter* ¡ max_iter **do**
8:             Further revise $\tilde{y}_{\mathrm{KG}}^+$ using error analysis
9:             Generate $y_{\mathrm{Gen}}$ with $\mathrm{LLM}_{\mathrm{Gen}}$
10:         **end while**
11:         **if** $y_{\mathrm{Gen}} = y_{\mathrm{gnd}}$ **then**
12:             Add $(y_{\mathrm{KG}}^+, y_{\mathrm{KG}}^-)$ as a contrastive DDPO training sample pair
13:         **else**
14:             Discard the sample
15:         **end if**
16:     **end if**
17: **end if**

---

The resulting framework effectively bridges the gap between initial model outputs and high-quality training samples while preserving the semantic-structural integrity of knowledge representations.

## 3.3 OPTIMIZATION APPROACH

Direct Preference Optimization (DPO) Rafailov et al. (2024) provides an effective framework for aligning language models with human preferences through pairwise comparisons. While theoretically sound, the method suffers from sparse supervision signals when applied to complex knowledge organization tasks. The method's uniform token-weighting scheme fails to prioritize structurally important segments in complex outputs like knowledge graphs, where discriminative signals often concentrate in specific entity relationships. Furthermore, the reward maximization objective can artificially amplify differences between semantically similar pairs, simultaneously depressing their probabilities and destabilizing training.

To address these issues, we adopted a variant of the DPO algorithm proposed Yu et al. (2024), known as Dense Direct Preference Optimization (DDPO). To make things clearer, we briefly review the DPO algorithm. The reward function for a specific output $y$ of input $x$ is represented as follows:

$$r(x, y) = \beta \log \frac{\pi_*(y \mid x)}{\pi_{\mathrm{ref}}(y \mid x)} + \beta \log Z(x) \tag{2}$$

Where $\beta$ is a constant hyperparameter, and $Z(x)$ is a partition function. $\pi_{ref}(y|x)$ is the base model we want to optimize, and kept fixed during training. $\pi_*(y|x)$ is the model we actually updated. Then we can get the DPO loss:

$$\begin{aligned} \mathcal{L}_{\mathrm{DPO}} &= -\mathbb{E}_{(x,y^+,y^-)} \left[ \log \sigma \Big( r(x, y^+) - r(x, y^-) \Big) \right] \\ &= -\mathbb{E}_{(x,y^+,y^-)} \left[ \log \sigma \left( \beta \left( \log \frac{\pi_*(y^+|x)}{\pi_{\mathrm{ref}}(y^+|x)} - \log \frac{\pi_*(y^-|x)}{\pi_{\mathrm{ref}}(y^-|x)} \right) \right) \right] \end{aligned} \tag{3}$$

DPO algorithm treats different token with uniform weight, and the score can be calculated as follow:

$$\log \pi(y \mid x) = \sum_{y_t \in y} \log p(y_t \mid x, y_{<t}) \tag{4}$$

$y_t$ is the $t$-th token of the response $y$. To ensure that the model pays more attention to the modified tokens $y_c$ compared to the unmodified tokens $y_u$ during optimization, we introduce a weighting

mechanism that assigns higher weights to the $y_c$ tokens when computing the score. The modified score calculation formula is as follows:

$$\log \pi(y \mid x) = \sum_{y_t \in y_u} \log p(y_t \mid x, y_{<t}) + \gamma \sum_{y_t \in y_c} \log p(y_t \mid x, y_{<t}) \tag{5}$$

$\gamma$ is a hyperparameter utilized to modulate the weight of tokens within $y_c$ when computing the score, while the weight of tokens in $y_u$ remains constant at 1.

While DDPO demonstrates effectiveness in our experiment, we observe that the simultaneous reward reduction phenomenon persists in certain cases. To mitigate this effect, we integrate supervised fine-tuning (SFT) loss as a regularization term, following recent theoretical insights Liu et al. (2024b). The complete loss function combines these components as follows:

$$\mathcal{L} = \mathcal{L}_{\text{DPO}} - \alpha \sum_{t=1}^{T} \log p(y_t \mid x, y_{<t}) \tag{6}$$

$\alpha$ is a hyperparameter that governs the weighting of the SFT loss within the overall loss function.

## 4 EXPERIMENTAL SETTINGS

In this section we first introduce our experimental settings, including datasets, evaluation metrics, and implementation details.

**Dataset**. In our experiments, we employed the challenging multi-hop question-answering dataset, MusiqueTrivedi et al. (2022b), to construct our training data. Following the approach of FlashRAGJin et al. (2024), we utilized Wikipedia as the retrieval document source. For both training and testing across various datasets, the bge-large-en-v1.5Xiao et al. (2023) model was used as the retrieval engine.

By leveraging the training data construction process described in Section 3.2, we generated 2,401 training data pairs from the 19,938 entries in the training set. For testing purposes, we utilized the Musique Development set. Additionally, to evaluate the model's out-of-distribution (OOD) performance, we encompasses both question answering and other knowledge-intensive tasks for testing. The QA benchmarks include Natural QuestionsKwiatkowski et al. (2019) and PopQAMallen et al. (2022) for single-hop retrieval evaluation, complemented by HotpotQAYang et al. (2018) for multi-hop reasoning assessment. To examine broader capabilities, we incorporate TruthfulQALin et al. (2021) for multiple-choice question answering and Zero-shot RELevy et al. (2017) for slot-filling. This multifaceted approach enables systematic analysis of the model's generalization abilities across varying task formats and difficulty levels while maintaining focus on core knowledge utilization challenges.

**Evaluation**. Following FlashRAGJin et al. (2024), we employ exact match (EM) and F1 score as evaluation metrics for the question answering task. For TruthfulQA evaluation, we utilize BLEU and ROUGE-1 scores. In the case of zero-shot relation extraction, we implement F1 score and precision as our evaluation metrics.

**Baseline**. We establish three baselines for comprehensive comparison. We first constructed a Vanilla RAG Pipeline, where the model directly generates answers from retrieved documents via in-context learning. RA-DiT is emplemented as a baseline, which trains the RAG system using the instruct-tuning method. DDR is also compared, a DPO-based approach that similarly focuses on optimizing the RAG system. In our experiments, we reimplement both RA-DiT and DDR using our reconstructed dataset. Details on construction and training can be found in Appendix A.2. To ensure a fair comparison, we exclusively train the generation module while keeping all other RAG components frozen during training.

**Implementation Details**. In our experiment, we employ Llama-3.1-8B-InstructTouvron et al. (2023) as the backbone models to construct most of the experiments. Training codes are modified from TRLvon Werra et al. (2020). During the retrieval phase, we retrieved five relevant documents for each query to construct the dataset. As for data construction, we utilized gpt-4o-miniAchiam et al. (2023) as the expert model to refine negative examples into positive ones. In the training

| Method | In Domain | | Out Of Domain | | | | | | | | | |
|---|---|---|---|---|---|---|---|---|---|---|---|---|
| | Musique | | NQ | | HotpotQA | | PopQA | | TruthfulQA | | Zero-shot RE | |
| | EM | F1 | EM | F1 | EM | F1 | EM | F1 | BLEU | Rouge-1 | F1 | Precision |
| *Llama-3.2-3B-Instruct* | | | | | | | | | | | | |
| Vanilla RAG | 4.47 | 9.89 | 33.2 | 45.59 | **27.89** | 37.75 | 34.32 | 41.6 | 6.36 | 13.78 | 48.7 | 47.4 |
| RA-DiT | 5.46 | 13.16 | 26.25 | 37.08 | 21.74 | 31.74 | 33.01 | 37.76 | 3.06 | 10.24 | 32.05 | 31.82 |
| DDR | **6** | **16.18** | 28.93 | 45.31 | 22.88 | 38.04 | 33.21 | 42.91 | 2.83 | 10.09 | **52.55** | **50.76** |
| KARE | 5.59 | 13.28 | **33.21** | **46.56** | 27.62 | **39.59** | **38.29** | **45.52** | **7.58** | **17.75** | 51.94 | 49.03 |
| *Llama-3.1-8B-Instruct* | | | | | | | | | | | | |
| Vanilla RAG | 6 | 12.37 | 34.64 | 48.3 | 29.52 | 40.66 | 35.43 | 44.1 | 5.43 | 15.03 | 51.62 | 49.3 |
| RA-DiT | **8.11** | 17.83 | 33.08 | 45.66 | 27.17 | 39.33 | 36.36 | 43.78 | 5.69 | 14.7 | 45.16 | 44.04 |
| DDR | 7.82 | **19.18** | 34.78 | 48.85 | 30.58 | 42.59 | 40.63 | 45.9 | 1.68 | 8.41 | 55.34 | 49.12 |
| KARE | 8.02 | 15.75 | **37.86** | **50.84** | **32.36** | **44.29** | **40.88** | **47.77** | **7.42** | **17.76** | **59.52** | **58.0** |
| *Qwen2.5-14B-Instruct* | | | | | | | | | | | | |
| Vanilla RAG | 6.66 | 14.58 | 32.67 | 47.01 | 30.64 | 42.31 | 36.38 | 43.98 | 6.9 | 18.15 | 53.26 | 49.27 |
| RA-DiT | 9.06 | 18.47 | 33.36 | 43.62 | 31.02 | 41.58 | 37.32 | 44.91 | 1.59 | 9.63 | 54.36 | **53.91** |
| DDR | **9.68** | **19.84** | 32.62 | 45.85 | 32.05 | 44.65 | **41.08** | 45.87 | 1.14 | 8.3 | 52.08 | 50.13 |
| KARE | 8.32 | 17.48 | **36.19** | **49.96** | **33.96** | **46.45** | 38.58 | **46.09** | **8.8** | **20.39** | **54.47** | 50.55 |

Table 1: Overall Performance of KARE-RAG comparing to different baseline methods. The **best** result in each block is highlighted. Experiments are performed on three instruction-tuned models spanning a range of parameter scales: Llama-3.2-3B-Instruct, Llama-3.1-8B-Instruct, and Qwen2.5-14B-Instruct.

process, we set the learning rate to 5e-5 and trained for one epoch. For the DPO training, $\beta$ was configured at 0.1, $\alpha$ was set to 0.1. Additionally, within the DDPO framework, the gamma parameter was established at 1.1. Besides we also conducted partial testing and comparative analysis of our method on the Qwen2.5-14B-InstructYang et al. (2024)Team (2024) and Llama-3.2-3B-InstructTouvron et al. (2023) models to validate the efficacy of our approach on different model size. For all optimization we use LoRAHu et al. (2021) for efficient training.

## 5 EVALUATION RESULTS

In this section, we first compare the overall performance of our method with several baseline models. Subsequently, we conduct ablation experiments and analyses to further validate the effectiveness of our approach.

### 5.1 MAIN RESULTS

The comprehensive experimental results presented in Table 1 demonstrate the robust performance of our approach across multiple evaluation dimensions. Our analysis reveals that DPO-based methods generally outperform SFT-based approaches, confirming that DPO is better suited for optimizing RAG systems. Most notably, KARE-RAG exhibits superior out-of-distribution generalization capabilities compared to all baseline methods, achieving an average EM score improvement of 4% across NQ, HotpotQA, and PopQA tasks for all three model sizes. These consistent improvements across diverse task formats and model scales are particularly remarkable given the limited quantity and diversity of training data. Furthermore, these benefits are achieved without compromising in-domain task performance, where KARE-RAG remains competitive with state-of-the-art baselines. The method's effectiveness scales consistently across different model architectures (from 3B to 14B parameters), demonstrating broad applicability and suggesting that the learned knowledge utilization patterns transfer effectively across model capacities. These findings collectively validate our approach's ability to develop robust knowledge processing strategies that generalize beyond specific training.

### 5.2 ANALYSIS

In this section, we conducted ablation experiments and analyses to further validate the effectiveness of our method.

**Impact of Different Knowledge Organization Formats**. We compare the effects of various intermediate information organization formats on training effectiveness to emphasize the importance

| Format | In Domain | | Out Of Domain | | | | | | | | | |
|---|---|---|---|---|---|---|---|---|---|---|---|---|
| | Musique | | NQ | | HotpotQA | | PopQA | | TruthfulQA | | Zero-shot RE | |
| | EM | F1 | EM | F1 | EM | F1 | EM | F1 | BLEU | Rouge-1 | F1 | Precision |
| ⌐ | 6.0 | 12.37 | 34.64 | 48.3 | 29.52 | 40.66 | 35.43 | 44.1 | 5.43 | 20.38 | 51.62 | 49.3 |
| Summary | 5.67 | 14.2 | 32.77 | 47.11 | 24.43 | 37.44 | 39.22 | 46.72 | **7.75** | **21.81** | 51.71 | 47.47 |
| Keypoints | 7.36 | **16.31** | 34.73 | 49.06 | 30.07 | 42.93 | **41.24** | **48.38** | 6.75 | 19.46 | 55.01 | 51.74 |
| Graph | **8.02** | 15.75 | **37.86** | **50.84** | **32.36** | **44.29** | 40.88 | 47.77 | 7.42 | 17.76 | **59.52** | **58.0** |

Table 2: Ablation Study. The test of different knowledge organization format under Vanilla RAG Pipeline. Experiment is conducted with model Llama-3.1-8B-Instruct

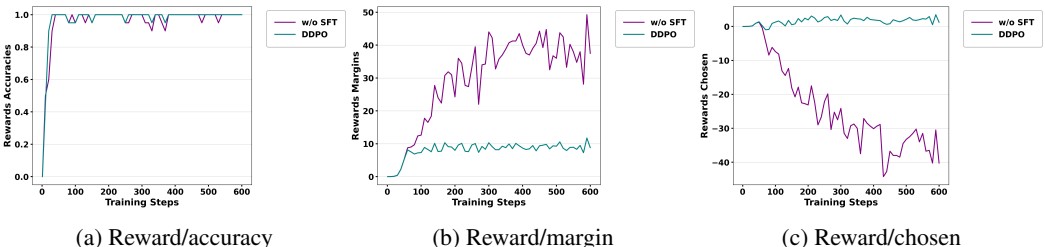

(a) Reward/accuracy        (b) Reward/margin        (c) Reward/chosen

Figure 3: Training curves with or without SFT Loss.

of structured representation. Experiments on Llama-3.1-8B-Instruct were conducted using two additional formats: a semi-structured keypoints format (one key point per line) and an unstructured summary format (organizing information into a summary). Specific prompts are provided in the Appendix A.6

As illustrated in Table 2, the structured graph representation leads to substantial improvements in both in-domain and out-of-domian tests, while the semi-structured keypoints format shows moderate gains. The unstructured summary format even shows a decline in performance after training. The results strongly suggest that structural inductive biases should be incorporated into RAG training objectives when possible, as they provide clearer learning signals for knowledge-intensive tasks while maintaining compatibility with standard inference pipelines.

**Importance of Token-Level Weighting**. The better performance of DDPO compared to DPO as shown in Table 3 validates the importance of our token-level weighting mechanism. Due to the extended length of our training outputs and the subtle distinctions between positive and negative examples, the standard DPO approach - which applies uniform weighting across all tokens - makes it particularly challenging for the model to learn meaningful patterns effectively. DDPO, on the other hand, enables more precise focus on discriminative features between positive and negative examples, leading to measurable improvements in model performance during inference. The consistent gains across different architectures and task domains further confirm the effectiveness of this design choice.

**The needs for SFT Loss** As shown in Table 3, incorporating both DPO Loss and SFT Loss significantly improves the model's training effectiveness and stability on OOD tasks, compared to using DPO Loss alone. Standard DPO Loss amplifies the reward difference between positive and negative examples, increasing the likelihood of sampling positive examples (Equation 3). Figures 3a and 3b show that standard DPO behaves as expected, with reward/accuracy approaching 1 and reward/margin increasing. However, due to the minimal differences between positive and negative examples and the lack of regularization in DPO, the model often overfits, causing both reward/chosen and reward/rejected to decrease (Figure 3c). This indicates that the model outputs fewer positive examples than the untrained model, leading to suboptimal training. Including SFT Loss mitigates this issue, keeping reward/chosen positive, maintaining reward/accuracy near 1, and preventing overfitting.

**Test General Ability**. While our training method was optimized for the RAG scenario, potentially affecting the model's general capabilities, we evaluated its performance on the MMLU Hendrycks et al. (2020) and MMLU-Pro Wang et al. (2024) datasets. As shown in Table 4, the training method did not negatively impact the model's general abilities. The MMLU metrics were consistent with the untrained model, and the MMLU-Pro metrics showed improvement. This suggests that our method

| Train Method | In Domain | | Out Of Domain | | | | | | | | | |
|---|---|---|---|---|---|---|---|---|---|---|---|---|
| | Musique | | NQ | | HotpotQA | | PopQA | | TruthfulQA | | Zero-shot RE | |
| | EM | F1 | EM | F1 | EM | F1 | EM | F1 | BLEU | Rouge-1 | F1 | Precision |
| *Llama-3.2-3B-Instruct* | | | | | | | | | | | | |
| ⌣ | 4.47 | 9.89 | 33.2 | 45.59 | **27.89** | 37.75 | 34.32 | 41.6 | 6.36 | 13.78 | 48.7 | 47.4 |
| SFT | 2.73 | 7.29 | 20.06 | 30.46 | 21.73 | 31.35 | 29.52 | 35.8 | 5.96 | 14.02 | 25.32 | 22.83 |
| DDPO | **5.59** | 13.28 | **33.21** | **46.56** | 27.62 | **39.59** | **38.29** | **45.52** | 7.58 | 17.75 | **51.94** | **49.03** |
| - w/o SFT Loss | 4.1 | **13.99** | 22.32 | 38.37 | 23.13 | 37.5 | 31.91 | 41.57 | 7.96 | **20.64** | 49.36 | 44.64 |
| - w/o Token Weight | 4.63 | 13.97 | 26.4 | 41.95 | 24.27 | 38.53 | 33.32 | 42.87 | **7.97** | 19.16 | 51.2 | 48.32 |
| *Llama-3.1-8B-Instruct* | | | | | | | | | | | | |
| ⌣ | 6 | 12.37 | 34.64 | 48.3 | 29.52 | 40.66 | 35.43 | 44.1 | 5.43 | 15.03 | 51.62 | 49.3 |
| SFT | 4.88 | 10.46 | 33.62 | 44.1 | 23.69 | 32.77 | 35.88 | 40.73 | 3.12 | 9.63 | 47.5 | 45.05 |
| DDPO | **8.02** | **15.75** | **37.86** | **50.84** | **32.36** | **44.29** | **40.88** | **47.77** | 7.74 | 17.76 | **59.52** | **58.0** |
| - w/o SFT Loss | 6.54 | 13.92 | 33.08 | 46.74 | 27.51 | 40.43 | 37.18 | 44.34 | 6.34 | **18.54** | 54.36 | 51.92 |
| - w/o Token Weight | 7.73 | 15.75 | 37.4 | 50.57 | 32.26 | 43.83 | 40.27 | 47.39 | **7.64** | 17.0 | 58.72 | 57.1 |
| *Qwen2.5-14B-Instruct* | | | | | | | | | | | | |
| ⌣ | 6.66 | 14.58 | 32.67 | 47.01 | 30.64 | 42.31 | 36.38 | 43.98 | 6.9 | 18.15 | 53.26 | 49.27 |
| SFT | 6.08 | 14.12 | 32.23 | 45.7 | 28.32 | 40.56 | 36.57 | 43.86 | 8.32 | 20.46 | 51.72 | 49.27 |
| DDPO | **8.32** | **17.48** | **36.19** | **49.96** | **33.96** | **46.45** | **38.58** | **46.09** | 8.8 | 20.39 | **54.47** | **50.55** |
| - w/o SFT Loss | 6.29 | 15.7 | 25.18 | 41.25 | 25.18 | 41.25 | 33.42 | 42.03 | 8.54 | **21.95** | 52.1 | 47.22 |
| - w/o Token Weight | 7.57 | 16.58 | 34.99 | 49.0 | 32.84 | 45.47 | 38.12 | 45.65 | **9.0** | 20.17 | 54.07 | 50.13 |

Table 3: Ablation Study. Comparative Analysis of Training Approaches: DDPO, SFT, and DDPO Variants (w/o SFT Loss, w/o Token Weighting). Here, We use the positive sample of each DPO pair for SFT training.

| Train Method | MMLU(Acc) | MMLU-Pro(Acc) |
|---|---|---|
| ⌣ | **68.0** | 43.97 |
| KARE-RAG | 67.9 | **44.60** |

Table 4: Test the general abblility of our method with MMLU and MMLU-Pro. Llama-3.1-8B-Instruct is applied for the test.

does not compromise general capabilities and may even improve performance on complex tasks in MMLU-Pro.

# 6 CONCLUSION

This paper presents KARE-RAG, a novel framework that enhances Retrieval-Augmented Generation systems through structured knowledge-aware training. Our method introduces an intermediate knowledge organization stage that explicitly models document utilization through structured representations, combined with an automated refinement pipeline that preserves structural consistency while correcting factual errors. The token-weighted DDPO optimization further enables focused learning on discriminative knowledge components. Experimental results demonstrate that this approach achieves significant improvements with limited training data while maintaining robust out-of-domain generalization. These benefits are accomplished without modifying standard RAG architectures or compromising general capabilities. The success of our method validates the importance of fine-grained training strategies for knowledge-intensive tasks and opens new directions for developing efficient and reliable RAG systems.

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

| Inference Pipeline | Train Method | In Domain Musique | | Out Of Domain NQ | | HotpotQA | | 2WikimultihopQA | | PopQA | | WebQ | | Average Gain | |
|---|---|---|---|---|---|---|---|---|---|---|---|---|---|---|---|
| | | EM | F1 | EM | F1 | EM | F1 | EM | F1 | EM | F1 | EM | F1 | EM | F1 |
| **Llama-3.2-3B-Instruct** | | | | | | | | | | | | | | | |
| Vanilla | ⌐ | 4.47 | 9.89 | 33.2 | 45.59 | **27.89** | 37.75 | 11.9 | 22.24 | 34.32 | 41.6 | 17.96 | 35.14 | | |
| | KARE-RAG | **5.59** | **13.28** | **33.21** | **46.56** | 27.62 | **39.59** | **15.5** | **25.53** | **38.29** | **45.52** | **19.44** | **36.97** | 1.76 | 2.37 |
| Knowledge Aware | ⌐ | 4.39 | 10.7 | 25.57 | 37.33 | 21.34 | 31.76 | 15.6 | 21.92 | 32.27 | 38.46 | 17.42 | 35.2 | | |
| | KARE-RAG | **6.99** | **15.8** | **28.51** | **40.72** | **28.85** | **38.79** | **26.67** | **33.45** | **36.96** | **42.24** | **18.65** | **36.68** | 5.49 | 5.44 |
| Chain Of Thought | ⌐ | 5.01 | 11.8 | 28.03 | 41.01 | 24.0 | 35.43 | 14.15 | 21.07 | 32.92 | 39.99 | **16.98** | 34.98 | | |
| | KARE-RAG | **7.24** | **16.8** | **29.31** | **42.73** | **26.41** | **39.03** | **22.57** | **30.69** | **36.23** | **43.42** | 16.14 | **35.64** | 2.92 | 3.81 |
| Chain Of Note | ⌐ | 4.18 | 10.63 | 29.15 | 42.42 | 22.04 | 32.84 | 10.19 | 17.43 | 30.83 | 38.34 | 15.94 | 34.59 | | |
| | KARE-RAG | **6.21** | **15.16** | **30.44** | **44.45** | **25.06** | **37.87** | **16.87** | **25.8** | **34.69** | **42.71** | 15.94 | **35.42** | 2.97 | 4.13 |
| **Llama-3.1-8B-Instruct** | | | | | | | | | | | | | | | |
| Vanilla | ⌐ | 6 | 12.37 | 34.64 | 48.3 | 29.52 | 40.66 | 14.94 | 24.89 | 35.43 | 44.1 | 16.83 | 35.33 | | |
| | KARE-RAG | **8.02** | **15.75** | **37.86** | **50.84** | **32.36** | **44.29** | **21.35** | **30.11** | **40.88** | **47.77** | **19.83** | **37.89** | 4.18 | 3.52 |
| Knowledge Aware | ⌐ | 7.57 | 15.13 | 33.07 | 46.03 | 30.2 | 41.17 | 18.54 | 25.78 | 38.79 | 44.76 | 20.13 | 37.86 | | |
| | KARE-RAG | **11.42** | **20.53** | **36.21** | **49.36** | **33.98** | **46.01** | **30.61** | **37.92** | **41.92** | **47.55** | **22.1** | **39.95** | 4.82 | 5.04 |
| Chain Of Thought | ⌐ | 8.52 | 16.4 | 35.63 | 48.48 | 31.11 | 42.45 | 16.71 | 23.52 | 37.71 | 44.35 | 20.47 | 38.8 | | |
| | KARE-RAG | **10.34** | **19.28** | **37.22** | **50.28** | **34.4** | **46.82** | **23.88** | **31.5** | **40.38** | **46.96** | **22.0** | **39.63** | 3.25 | 3.52 |
| Chain Of Note | ⌐ | 6.83 | 14.47 | 35.06 | 48.61 | 29.05 | 40.7 | 17.37 | 24.2 | 36.06 | 43.32 | 19.44 | 38.7 | | |
| | KARE-RAG | **9.18** | **18.44** | **37.55** | **50.73** | **33.11** | **45.34** | **22.56** | **30.44** | **39.36** | **46.3** | **21.26** | **40.07** | 3.37 | 3.47 |
| **Qwen2.5-14B-Instruct** | | | | | | | | | | | | | | | |
| Vanilla | ⌐ | 6.66 | 14.58 | 32.67 | 47.01 | 30.64 | 42.31 | 19.73 | 29.5 | 36.38 | 43.98 | 18.36 | 35.11 | | |
| | KARE-RAG | **8.32** | **17.48** | **36.19** | **49.96** | **33.96** | **46.45** | **25.14** | **33.72** | **38.58** | **46.09** | **20.03** | **36.95** | 3.22 | 3.05 |
| Knowledge Aware | ⌐ | 9.02 | 17.67 | 36.75 | 50.15 | 35.91 | 48.25 | 24.51 | 30.88 | 39.2 | 44.95 | 19.88 | 37.82 | | |
| | KARE-RAG | **11.09** | **20.78** | **37.19** | **50.46** | **37.49** | **50.56** | **34.65** | **42.12** | **40.65** | **47.3** | **20.72** | **38.95** | 2.89 | 3.27 |
| Chain Of Thought | ⌐ | 9.02 | 17.04 | **37.99** | **51.16** | 34.8 | 46.66 | 21.15 | 28.08 | 38.28 | 45.35 | **19.78** | **37.48** | | |
| | KARE-RAG | **10.72** | **21.63** | 37.34 | 51.07 | **36.88** | **50.22** | **29.61** | **38.37** | **39.42** | **46.87** | 18.8 | 37.36 | 2.01 | 3.03 |
| Chain Of Note | ⌐ | 8.36 | 16.81 | 35.31 | 49.25 | 32.92 | 45.54 | 21.37 | 30.61 | 36.31 | 44.26 | 17.77 | 36.2 | | |
| | KARE-RAG | **9.27** | **19.55** | **36.49** | **50.49** | **35.04** | **48.39** | **28.05** | **37.61** | **37.99** | **45.65** | **18.36** | **37.58** | 2.45 | 2.77 |

Table 5: The results demonstrate the effectiveness of the KARE-RAG-trained model across various RAG generation pipelines, with the final two columns showing its average EM and F1 score improvements over the baseline (untrained) model for each pipeline.

# A  APPENDIX

## A.1  LLM EMPLOYMENT

In the preparation of this paper, we employed large language models (LLMs) exclusively to assist with language-related aspects of the writing process. Specifically, LLMs were used to refine grammatical structures, facilitate translation of certain text segments, and provide final polishing of the manuscript to improve clarity and readability. No substantial intellectual, methodological, or analytical input was contributed by LLMs, and all scientific content, results, interpretations, and conclusions remain entirely our own.

## A.2  ADDITIONAL BASELINE SETTINGS

In this section, we detail the data construction process for the DDR and RA-DiT baseline methods in our experiments. To ensure consistency in training data, we follow the methodologies outlined in DDRLi et al. (2024a) and RA-DiTLin et al. (2023) to construct our dataset using Musique. For RA-DiT, we employ the same instruction prompts as described in the original paper and maintain a fixed retrieval size of five documents per query. For DDR, we provide retrieved documents to the large language model and conduct multi-round sampling with varying temperature settings, adhering to the approach proposed in the DDR paper. The sampling process is jointly controlled by F1 and EM metrics: examples are considered positive if either F1 or EM exceeds a predefined threshold, while those falling below thresholds on both metrics are treated as negative samples. Cases failing to yield valid positive or negative examples are discarded. Finally, we compose DPO training pairs by selecting samples with the highest and lowest metric scores respectively, which constitute our final training data.

For training both RA-DiT and DDR, we employ the training methodology provided by TRLvon Werra et al. (2020), utilizing a learning rate of 5e-5 for one complete epoch. In the case of DDR, we set the $\beta$ parameter to 0.1 during training. All training procedures are implemented using LoRAHu et al. (2021) to reduce computational overhead while maintaining alignment with our proposed approach. This configuration ensures consistent training efficiency while preserving the comparative validity with our method.

| Train Method | In Domain | Out Of Domain | | | | | |
| --- | --- | --- | --- | --- | --- | --- | --- |
| | Musique ACC | NQ ACC | HotpotQA ACC | 2WikimultihopQA ACC | PopQA ACC | WebQ ACC | Average Gain ACC |
| Llama-3.2-3B-Instruct | | | | | | | |
| ⟍ | 10.92 | 52.32 | 37.58 | 35.90 | 48.07 | 45.62 | |
| KARE-RAG | **15.68** | **56.22** | **43.36** | **38.14** | **53.80** | **51.62** | 4.74 |
| Llama-3.1-8B-Instruct | | | | | | | |
| ⟍ | 7.70 | 46.21 | 32.22 | 29.80 | 41.57 | 39.61 | |
| KARE-RAG | **14.93** | **57.45** | **45.05** | **38.48** | **54.32** | **50.49** | 10.6 |
| Qwen2.5-14B-Instruct | | | | | | | |
| ⟍ | 7.78 | 52.15 | 30.83 | 35.42 | 45.29 | 46.65 | |
| KARE-RAG | **9.76** | **55.57** | **35.84** | **35.56** | **48.21** | **49.41** | 2.71 |

Table 6: Performance Evaluation on IRCOT Pipeline. Results demonstrate KARE-RAG's effectiveness in multi-step retrieval scenarios across different model scales. The **best** result in each model group is highlighted. The rightmost column indicates average accuracy gains over baseline models.

## A.3   COMPUTATIONAL EFFICIENCY

Our framework demonstrates strong practical efficiency in both data construction and model training phases. For data generation, the pipeline leverages GPT-4o-mini's capability to perform targeted refinements, keeping the construction cost around $20. The training process employs parameter-efficient LoRA adapters combined with DeepSpeed-Zero3 optimization, enabling efficient fine-tuning on a single A800-40G GPU. Specifically, training completes in under 1 hour for 3B models and within 4 hours for 14B models.

## A.4   GENERALIZATION ACROSS DIFFERENT GENERATION PIPELINES

To further validate the versatility of our approach, we evaluated the trained models across multiple generation pipelines. These pipelines include the Knowledge-Aware Pipeline which has benn used for data construction, Chain-of-Thought pipelineWei et al. (2022), the Chain-of-Note pipelineYu et al. (2023), and more complex IRCOT pipelineTrivedi et al. (2022a)

The results can be viewed in Table 5, Table 6. While the most significant improvements naturally occur with the KA-Pipeline (aligned with our training methodology), all tested pipelines demonstrate consistent performance gains without degradation. This pattern holds particularly true for complex reasoning tasks, where our method shows stable improvements regardless of the specific pipeline architecture.

These findings strongly suggest that KARE-RAG's effectiveness stems from its fundamental enhancement of how models process and utilize retrieved information, rather than from optimization for any particular pipeline design. The consistent performance gains across diverse implementations highlight the method's general applicability to various RAG architectures in real-world scenarios.

## A.5   COMPARISON WITH ROBUST LLM TRAINING

In this section, we present a comparative analysis between our proposed method and existing Robust LLM training approaches, which enhance model resilience to noise by manipulating the retrieved documents used for generating final answers. Our experimental setup involves retrieving 10 candidate documents per training sample, which are categorized into three distinct types: (1) golden documents containing correct answers, (2) relevant noise comprising retrieved but non-answer-bearing texts, and (3) irrelevant noise consisting of randomly sampled documents from the corpus. We evaluate three robust training paradigms: (i) REALMLin et al. (2023), which exclusively uses golden documents for training; (ii) RetRobustYoran et al. (2023), which randomly selects one document from either golden, relevant noise, or irrelevant noise categories during training; and (iii) Golden + 3 Random, where each training sample combines the golden document with three randomly sampled documents. As demonstrated in the Table 7, our method outperforms these established robust training techniques, suggesting that training models to generate effective knowledge graphs represents a viable alternative approach for developing robust LLMs. This comparative study highlights the

| Method | In Domain | | Out Of Domain | | | | | | | | | |
|---|---|---|---|---|---|---|---|---|---|---|---|---|
| | Musique | | NQ | | HotpotQA | | PopQA | | TruthfulQA | | Zero-shot RE | |
| | EM(%) | F1(%) | EM(%) | F1(%) | EM(%) | F1(%) | EM(%) | F1(%) | BLEU(%) | Rouge-1(%) | F1(%) | Precision(%) |
| Vanilla RAG | 6 | 12.37 | 34.64 | 48.3 | 29.52 | 40.66 | 35.43 | 44.1 | 5.43 | 15.03 | 51.62 | 49.3 |
| Realm | 6.08 | 13.46 | 29.02 | 39.21 | 24.94 | 35.18 | 31.2 | 36.73 | 2.63 | 10.39 | 40.23 | 39.52 |
| RetRobust | 5.25 | 12.62 | 28.99 | 39.06 | 24.27 | 34.08 | 32.52 | 36.75 | 3.53 | 10.96 | 37.97 | 36.9 |
| Golden + 3 Random | 6.21 | 13.82 | 29.69 | 39.89 | 25.35 | 35.84 | 32.39 | 37.66 | 2.06 | 9.91 | 34.78 | 34.61 |
| KARE | **8.02** | **15.75** | **37.86** | **50.84** | **32.36** | **44.29** | **40.88** | **47.77** | **7.42** | **17.76** | **59.52** | **58.0** |

Table 7: Comparative Results Between Our Method and Robust LLM Training Using Llama-3.1-8B-Instruct

effectiveness of our knowledge graph-based methodology in enhancing model robustness against noisy inputs while maintaining generation quality.

## A.6 EXPERIMENT PROMPTS

In this section, we present all the prompts utilized in our experiments. For Vanilla RAG, we directly used the standard prompt template provided by FlashRAGJin et al. (2024) without any modifications. Table 8 contains the prompts for the Knowledge-Aware generation when employing a Knowledge Graph as the intermediate representation for knowledge organization. Tables 9 and 10 display the prompts for the scenarios using Keypoints and Summary, respectively. Table 11 showcases the prompts we employed during the Knowledge Graph refinement phase.

| | |
|---|---|
| **Knowledge-Aware** | |
| System Prompt | You are a helpful AI assistant that are good at extracting crucial information from documents which are helpful for answering a given question. For a given question, focus on identifying the key entities, their attributes, and their relationships that are directly relevant to generating an accurate answer. Only include entities and attributes that are crucial for understanding and forming the answer, and avoid unnecessary details. |
| | Your response **must include the following keys** and strictly **adhere to the exact structure** without any additional text before or after the keys: |
| | Entities:
- [Entity 1] (Attributes: [Attribute 1, Attribute 2, ...])
- [Entity 2] (Attributes: [Attribute 1, Attribute 2, ...])
... |
| | Relationships:
1. [Entity 1] → [Relationship] → [Entity 2]
2. ... |
| | **Important Note:**
1. **Do not include any extra commentary** or unnecessary details in the response.
2. Closely follow the structure provided above and ensure that the response is concise and directly addresses the question.
3. **Don't provide the answer to the question.** Instead, focus on extracting the key entities, attributes, and relationships that are essential for answering the question accurately. |
| User Prompt | Question: {question}
Documents: {reference}
Knowledge Graph: |
| **CoT** | |
| System Prompt | You are a helpful AI assistant that are good at doing reasoning on the knowledge graph to answer a given question. For a given question and relevant knowledge graph, provide the step by step reasoning process to derive the answer. Make sure the reasoning steps are logical, coherent, and directly related to the question and the information in the knowledge graph. |
| User Prompt | Question: {question}
Knowledge Graph: $\{y_{KG}\}$
Reasoning Steps: |
| **Generation** | |
| System Prompt | You are a helpful AI assistant that are good at generating final answers based on the reasoning steps provided for a given question. |
| | **Important Notes:**
1. Make sure the final answer is accurate, concise, and directly addresses the question.
2. Only give me the answer and **do not output any other words.** |
| User Prompt | Question: {question}
Reasoning Steps: $\{y_{CoT}\}$
Answer: |

Table 8: Knowledge-Aware Gneration Prompt(Graph Format)

| | |
|---|---|
| **Knowledge-Aware** | |
| System Prompt | You are a helpful AI assistant that are good at extracting crucial information from documents which are helpful for answering a given question.For a given question, please thoroughly analyze the list of documents (given as strings) and extract the most relevant key points that directly answer the question.Ensure that each key point is directly supported by evidence found within the documents, and avoid unnecessary details. |
| | Your response **must include the following keys** and strictly **adhere to the exact structure** without any additional text before or after the keys: |
| | Key Points: |
| | 1. [Key Point 1] |
| | 2. [Key Point 2] |
| | ... |
| | **Important Note:** |
| | 1. **Do not include any extra commentary** or unnecessary details in the response. |
| | 2. Closely follow the structure provided above and ensure that the response is concise and directly addresses the question. |
| | 3. **Don't provide the answer to the question.** Instead, focus on extracting the key points that are essential for answering the question accurately. |
| User Prompt | Question: {question} |
| | Documents: {reference} |
| | KeyPoints: |
| **CoT** | |
| System Prompt | You are a helpful AI asssitant that are good at doing reasoning on the keypoints to answer a given question.For a given question and relevant keypoints, provide the step by step reasoning process to derive the answer.Make sure the reasoning steps are logical, coherent, and directly related to the question and the information in the keypoints. |
| User Prompt | Question: {question} |
| | Keypoints: $\{y_{\text{Key}}\}$ |
| | Reasoning Steps: |
| **Generation** | |
| System Prompt | You are a helpful AI assistant that are good at generating final answers based on the reasoning steps provided for a given question. |
| | **Important Notes:** |
| | 1. Make sure the final answer is accurate, concise, and directly addresses the question. |
| | 2. Only give me the answer and **do not output any other words.** |
| User Prompt | Question: {question} |
| | Reasoning Steps: $\{y_{\text{CoT}}\}$ |
| | Answer: |

Table 9: Knowledge-Aware Gneration Prompt(Kepoints Format)

| Knowledge-Aware | |
| --- | --- |
| System Prompt | You are a helpful AI assistant that are good at extracting crucial information from documents which are helpful for answering a given question.For a given question, you need to extract a note that is directly relevant to generating an accurate answer.

**Important Note:**
1. **Don't directly provide the answer to the question.** Instead, focus on extracting the relevant information that is essential for answering the question accurately and formulating a note.
2. The length of the response is limited, so make sure to include only the most relevant information. |
| User Prompt | Question: {question}
Documents: {reference}
Note: |
| CoT | |
| System Prompt | You are a helpful AI assistant that are good at doing reasoning on the note to answer a given question.For a given question with relevant note, provide the step by step reasoning process to derive the answer.Make sure the reasoning steps are logical, coherent, and directly related to the question and the information in the note. |
| User Prompt | Question: {question}
Note: $\{y_{\text{Note}}\}$
Reasoning Steps: |
| Generation | |
| System Prompt | You are a helpful AI assistant that are good at generating final answers based on the reasoning steps provided for a given question.

**Important Notes:**
1. Make sure the final answer is accurate, concise, and directly addresses the question.
2. Only give me the answer and **do not output any other words.** |
| User Prompt | Question: {question}
Reasoning Steps: $\{y_{\text{CoT}}\}$
Answer: |

Table 10: Knowledge-Aware Gneration Prompt(Summary Format)

| Check Answerability | |
|---|---|
| System Prompt | You are a helpful AI assistant that is very good at judging whether the answer can be derived from the documents for a given question. You will be given a question, a set of documents, and golden answers. Please determine whether any of the golden answers can be derived from the documents for the given question.
If any of the golden answers can be derived from the documents, the judgement should be "True". If none of the golden answers can be derived from the documents, the judgement should be "False". Do not output any explanation, only output the judgement. |
| User Prompt | Question: {question}
Documents: {reference}
Golden Answers: {golden_answers}
Judgement: |
| Refine Prompt | |
| System Prompt | You are a professional AI assistant that is very good at refining the flawed knowledge graph. The knowledge graph is extracted from the documents for a given question so that it can be helpful for answering the question. But the current knowledge graph may contain incorrect information, redundant information, or lack critical information, making it impossible to deduce the correct answer for the given question. You will be given a question, a set of documents, the flawed knowledge graph and golden answers. Your task is to add, remove, or modify the content in the knowledge graph to make it accurate and relevant to answering the question. Make sure that the refined knowledge graph's content is both relevant to answering the question and entirely derived from the provided document.

Important Notes:
1. The new knowledge graph should be refined based on the flawed knowledge graph, **don't start from scratch**.
2. The refined knowledge graph should be of the same format as the flawed knowledge graph.
3. **Do not directly add the golden answers to the knowledge graph, the refined knowledge graph should be derived from the documents.**
4. Do not output the explanation of your changes, only output the refined knowledge graph! |
| User Prompt | Question: {question}
Documents: {reference}
Flawed Knowledge Graph: $\{y_{\mathrm{KG}}^{-}\}$
Golden Answers: {golden_answers}
Refined Knowledge Graph: |

Table 11: Refinement Prompts

