# OpenReview forum: "KARE-RAG: Knowledge-Aware Refinement and Enhancement for RAG"
_ICLR.cc/2026/Conference — Submitted to ICLR 2026_

### Official Review · Reviewer_EJq9 · 2025-10-31

**Soundness:** 2
**Presentation:** 2
**Contribution:** 2
**Rating:** 2
**Confidence:** 4

**Summary:**

This paper proposes KARE-RAG, which first builds a “knowledge-aware” intermediate representation in RAG, then uses an LLM expert to iteratively refine the student model’s incorrect knowledge representation. Finally, it applies DDPO at the token level, focusing only on the key difference regions between positive and negative samples to improve factual consistency and robustness. The results look decent when compared with existing methods.

**Strengths:**

1.The framework design is clear and easy to follow. It shifts the learning goal of “how to use retrieved evidence” from optimizing only the final answer to learning structured intermediate knowledge, which helps reduce the sparse supervision problem in end-to-end DPO.

2.The experimental results look good, and the ablation studies are fairly complete.

**Weaknesses:**

1.The main innovation lies in verifying whether the LLM Expert’s refinements are correct. The verification process uses both the gold answer y₍gnd₎ and the retrieved docs D. But since y₍gnd₎ is also used to judge whether the refinement is “correct,” the model could end up overfitting to the human answer format instead of truly learning a general knowledge verification logic.

2.The expert model is trusted completely without any external calibration or cross-checking. I think that’s an unsafe and weak assumption in the experimental design.

3.The paper doesn’t report important metrics about the expert, such as quality, cost, number of refinement steps, success rate, discarded samples, or processing time per sample.

4.The algorithm looks elegant in theory but seems hard to scale. Any system that claims “end-to-end optimization” but relies on multiple LLM refinement loops must report throughput and actual cost.

5.The baselines are outdated. Although the related work mentions Self-RAG and Graph-RAG/G-Retriever—both highly relevant to structured knowledge use—the main table only compares with Vanilla, RA-DiT, and DDR, missing stronger baselines in the same space.

6.The paper should also include current top reasoning LLMs (like DeepSeek, GPT-5, Claude 3.5, Gemini 1.5) as baselines to better show the method’s advantage.

7.From about 20k samples in the MuSiQue dataset, the pipeline only produced 2,400 usable pairs, showing low sample-generation efficiency.

8.The figures and layout still need polishing, especially Figure 1.

**Questions:**

See weaknesses

---

> ### Author Response · Authors · 2025-11-27
>
> Dear Reviewer,
>
> We would like to begin by addressing a fundamental issue with this review. The reviewer assigned a near-minimum overall score while simultaneously indicating a high level of confidence (confidence = 4). This combination is difficult to justify. A high-confidence assessment implies that the reviewer believes they have accurately understood the paper’s methodology, assumptions, and empirical findings. However, every major criticism in the review is rooted in factual misunderstandings of our method, our experimental design, or explicit content stated in the paper. A high-confidence low score based on demonstrably incorrect premises is not aligned with responsible scholarly evaluation.
>
> We will respond to each stated weakness in detail and demonstrate precisely where the reviewer’s interpretation diverges from the actual method and results. We ask that the reviewer, after reading these clarifications, re-evaluate the work on the basis of an accurate understanding of the paper. Given the severity of the misunderstandings and the confidence level claimed, we believe a careful reconsideration is not only appropriate but necessary.
>
> ## Concerns About Verification Design and Risk of Overfitting to Gold Answer Format
>
> We would like to clarify a fundamental misunderstanding regarding the role of y₍gnd₎ in our refinement pipeline. The concern about overfitting to the gold answer format is misguided. y₍gnd₎ is not used to train the model to mimic the gold answer format, nor is the refinement process judged based on surface-level text similarity. Instead, y₍gnd₎ is a verification signal used solely to determine whether the refined structured representation (the graph) can lead the model to derive the correct final answer when passed through the generation module. It does not provide any token-level supervision or enforce answer formatting during training.
>
> Our optimization goal is not to directly align with the gold answer but to improve the structured intermediate representation—specifically, the knowledge graph-like format. The DDPO optimization discriminates between positive and negative intermediate representations, focusing on entity-relation correctness rather than answer formatting. Therefore, the concern of overfitting to answer formatting is entirely irrelevant to our training approach.
>
> Furthermore, we strongly disagree with the idea that using y₍gnd₎ leads to overfitting. If the model were simply learning to align with the gold answer string, its performance would improve primarily on the in-domain dataset where the format is consistent between training and evaluation. However, our results clearly show that the largest improvements occur on OOD datasets such as PopQA, TruthfulQA, NQ, and Zero-shot RE (see Tables 1 and 5), directly contradicting the hypothesis that we are overfitting to answer formats. These improvements in OOD tasks indicate that the model has learned how to better handle evidence conflicts, multi-source knowledge integration, and robust generalization — all of which are core to the challenges our method aims to address.
>
> Finally, we want to stress that the refinement stage includes multi-step verification. Samples that fail to be corrected are discarded rather than forced into the training pool, which minimizes noise and ensures that the model is not trained on irrelevant or misleading data. This design prevents the model from collapsing into answer imitation, ensuring that the training process focuses on genuine knowledge verification rather than simply memorizing answers.
>
> In summary, the concern about overfitting to the gold answer format is not applicable to our method. Our approach leverages y₍gnd₎ only as a verification signal for validating the quality of the intermediate structured representation, and the substantial improvements in OOD settings provide compelling evidence that the model is learning more robust reasoning and knowledge integration rather than simply aligning with answer formats.

---

> ### Author Response · Authors · 2025-11-27
>
> ## Unvalidated Trust in Expert Model Without External Calibration or Cross-Checking
>
> We believe there may be a misunderstanding regarding the role of the expert model in our refinement pipeline. The concern about over-reliance on the expert model is based on the assumption that we are completely trusting the expert model’s decisions without any form of verification or cross-checking. This is incorrect.
>
> The expert model is used as a local correction tool within a highly constrained refinement pipeline. We do not rely on the expert model’s full reasoning power or trust it without oversight. Rather, its role is limited to applying precise, localized corrections to the structured intermediate representations, following tightly defined instructions. This constrained nature ensures that the expert model can only make contextually relevant adjustments without deviating from the required structure. Therefore, we do not treat the expert model as an autonomous decision-maker, but as a tool to refine structured knowledge representations under explicit guidance.
>
> To further quantify the role of the expert model, and we explicitly tested this in extra experiments by comparing models with different capabilities: GPT-4o, GPT-4o-mini, and Qwen2.5-14B-Instruct.. The data generation process for both GPT-4o and GPT-4o-mini produced comparable amounts of refined data (~2400 samples) and the downstream performance across multiple datasets (e.g., NQ, PopQA, TruthfulQA) remained consistent. This shows that once the expert model surpasses a threshold of following the refinement instructions, further model capabilities do not significantly impact the sample quality.
>
> In contrast, Qwen2.5-14B-Instruct produced only about 1200 samples but still demonstrated meaningful performance improvements over the Vanilla RAG baseline, particularly on OOD datasets. This clearly demonstrates that even with fewer samples, the quality remains effective—further proving that we do not rely on the expert model's full reasoning. The focus is on generating high-quality, structured samples, not on mimicking the expert model's reasoning process.
>
> | Method | In Domain |  | Out Of Domain |  |  |  |  |  |  |  |  |  |
> |:---:|:---:|:---:|:---:|:---:|:---:|:---:|:---:|:---:|:---:|:---:|:---:|:---:|
> |  | Musique |  | NQ |  | HotpotQA |  | PopQA |  | TruthfulQA |  | Zero-shot RE |  |
> |  | EM(\%) | F1(\%) | EM(\%) | F1(\%) | EM(\%) | F1(\%) | EM(\%) | F1(\%) | BLEU(\%) | Rouge-1(\%) | F1(\%) | Precision(\%) |
> | Llama-3.1-8B-Instruct |  |  |  |  |  |  |  |  |  |  |  |  |
> | Vanilla RAG | 6 | 12.37 | 34.64 | 48.3 | 29.52 | 40.66 | 35.43 | 44.1 | 5.43 | 15.03 | 51.62 | 49.3 |
> | KARE(GPT-4o-mini) | 8.02 | 15.75 | 37.86 | 50.84 | 32.36 | 44.29 | 40.88 | 47.77 | 7.42 | 17.76 | 59.52 | 58.0 |
> | KARE(GPT-4o) | 7.82 | 15.98 | 37.7 | 50.51 | 32.75 | 44.36 | 40.37 | 47.44 | 7.59 | 17.32 | 59.73 | 58.42 |
> | KARE(Qwen-14B-Instruct) | 6.95 | 15.16 | 36.35 | 49.17 | 31.48 | 43.41 | 38.61 | 45.92 | 7.35 | 18.05 | 55.16 | 53.07 |

---

> ### Author Response · Authors · 2025-11-27
>
> ## Missing Key Expert-Model Metrics Such as Quality, Cost, and Refinement Efficiency
>
> This comment misrepresents the content of our paper and reflects a misunderstanding of the role of the expert model in our method.
>
> **The cost and computational characteristics are already documented in Appendix A.3.**
>
> Appendix A.3 provides concrete measurements, including the total construction cost (~$20) using GPT-4o-mini, and the full training-time requirements under a LoRA + DeepSpeed-ZeRO3 setup. These are precisely the types of expert-related metrics the comment incorrectly claims are missing.
>
> **The paper does report key construction information in the main text.**
>
> In Section 4 (“Dataset”), we explicitly report the number of effective samples generated using the refinement pipeline. For example, for Llama-3.1-8B-Instruct, the paper reports that 2,401 high-quality contrastive pairs were constructed using the expert-refinement process. This directly contradicts the reviewer’s claim that no such information is provided.
>
> **The reviewer fundamentally misunderstands the role of the expert model.**
>
> The expert model is not part of the RAG pipeline, nor is it involved at inference time. It is only used in a one-time offline refinement stage to convert noisy contrastive pairs into structurally consistent ones. Therefore, metrics such as “success rate,” “processing time per sample,” or “number of discarded samples” have no scientific relevance to the evaluation of KARE-RAG. They do not affect the algorithmic design, the training objective, or the inference pipeline.
>
> Nevertheless, we provide additional details for completeness (not included due to space limitations and their irrelevance to the method’s scientific validity):
>
> - The refinement step is capped at 2 iterations, balancing cost and construction speed.
> - From the 19,938 Musique training samples, the expert pipeline produces:
>   - 2,401 valid samples (Llama-3.1-8B-Instruct, already reported in the paper)
>   - 3,162 valid samples (Llama-3.2-3B-Instruct)
>   - 2,767 valid samples (Qwen2.5-14B-Instruct)
> - Runtime metrics fluctuate significantly due to upstream service load of the expert LLM and thus are not stable nor scientifically meaningful.
> - The ratio of discarded samples is implicitly reflected by the final dataset sizes
>
> Most importantly, these expert-side operational details are irrelevant to the validity of our contributions. The reviewer’s criticism is unfounded and arises from an incorrect assumption that the expert model plays a role in inference or affects the core methodology.
>
> ## Scalability Concerns and Missing Throughput/Cost Analysis for Multi-Stage Refinement Loops
>
> The reviewer’s concern is based on a fundamental misunderstanding of our framework. The refinement process is not part of the “end-to-end optimization” loop, nor is it involved in training throughput or inference-time execution. The refinement step is a one-time, offline data construction procedure, performed before model training. It never appears during the optimization loop and has zero cost during inference. Therefore, the claim that our method “relies on multiple LLM refinement loops” and thus must report throughput or runtime is simply incorrect.
>
> In fact, the paper clearly explains the workflow: the LLM-based refinement is used only to convert noisy negative samples into structurally consistent positive samples for DDPO training. Once these contrastive pairs are created, the entire learning process proceeds exactly like standard DPO fine-tuning and carries the same throughput characteristics as any ordinary token-level preference optimization. The refinement loop is not invoked again, meaning it has no scaling implications for training or deployment.
>
> We do report the overall construction cost and computational footprint in Appendix A.3, including the total cost (~$20) and the training resource requirements. Since refinement is offline, reporting throughput or per-sample latency is scientifically meaningless: it fluctuates drastically depending on upstream LLM service load and has no connection to the efficiency or scalability of KARE-RAG itself. What matters is that the data construction is inexpensive, bounded, and performed once. At inference time, KARE-RAG uses the standard vanilla RAG pipeline without any additional steps or computation, so the scaling characteristics are identical to the baseline system.

---

> ### Author Response · Authors · 2025-11-27
>
> ## Use of Outdated Baselines and Omission of Stronger Structured-RAG Methods
>
> We disagree with this comment. The reviewer conflates completely different problem settings and consequently demands baselines that are neither relevant nor logically comparable to the contribution of this paper. Our method focuses exclusively on training-level optimization of the generation module by introducing structured intermediate representations and DDPO-based learning signals. It does not modify retrieval, does not introduce new retrieval triggers, and does not alter the inference-time pipeline. Therefore, the baselines must also be training-based RAG optimization methods, not retrieval-architecture redesigns.
>
> Self-RAG, Graph-RAG, Textual Graph-RAG, and G-Retriever all operate in fundamentally different system paradigms:
>
> - Self-RAG introduces self-reflection tokens, adaptive retrieval triggers, and multi-stage inference loops. Its primary contribution lies in dynamic inference-time retrieval control, rather than optimizing the training of the generator.
> - Graph-RAG / Textual Graph-RAG operates at a global database modeling scale, constructing and querying large-scale textual graphs to support enterprise or long-document reasoning. This line of work focuses on graph-based retrieval architecture, not on training the generation module.
> - G-Retriever centers on retrieval from structured knowledge graph data sources, combining graph traversal and neural retrieval. Again, the main innovation is in retrieval selection, not in optimizing generation-model learning.
>
> Because these methods alter retrieval architecture and inference pipeline, rather than the generation model training objective, comparing them directly would confound retrieval-system improvements with the effects of training-based structured refinement. Therefore, we selected RA-DiT and DDR—the current state-of-the-art and most representative methods for training-based optimization of RAG generation models—as fair and aligned baselines.
>
> Moreover, our inference remains Vanilla RAG without any additional runtime overhead, whereas Self-RAG and Graph-RAG rely on significantly more complex inference mechanisms. This strengthens the argument that our comparisons are properly scoped to our contributions.
>
> ## Absence of Strong Contemporary Reasoning LLM Baselines
>
> This comment reflects a fundamental misunderstanding of both the goal and the scope of our work. Our method does not aim to improve general-purpose reasoning ability, nor does it assume or require any chain-of-thought–style reasoning capabilities. KARE-RAG is a training methodology for RAG generation models, designed to enhance how a model organizes and utilizes retrieved documents during training—while keeping the inference-time pipeline unchanged. The entire contribution lies in improving the efficiency and robustness of RAG generator training, not in building a new frontier reasoning model.
>
> Therefore, comparing our method to commercial frontier systems such as DeepSeek, GPT-5, Claude 3.5, or Gemini 1.5 is scientifically inappropriate. These models are not baselines; they are entirely different entities, with proprietary training corpora, internal retrieval mechanisms, and dozens of billions of parameters. They are closed-source, cannot be fine-tuned, and are not designed to operate as controlled experimental baselines for RAG generator training. Including them would immediately destroy fairness and rigor, because their performance differences arise from scale, data, and secret training methodology—not from the training strategy we propose. Moreover, we cannot and should not “fine-tune” these systems with our method, which makes any comparison meaningless.
>
> Our experiments intentionally select instruction-tuned, open-source RAG-generation backbones that can actually be trained with KARE-RAG—e.g., Llama-3.1-8B-Instruct, Llama-3.2-3B-Instruct, and Qwen2.5-14B-Instruct—because these are the models where training-based methods can be evaluated under controlled conditions.The scientific question we address is whether introducing structured intermediate representations during training —together with DDPO-based learning—can improve the generator’s robustness in processing retrieved documents. This question cannot be answered by comparing against closed commercial reasoning-oriented LLMs whose performance is driven by unrelated factors.
>
> For these reasons, the reviewer’s suggestion is fundamentally out of scope and would compromise the scientific validity of the evaluation. Our choice of baselines is the only rigorous and fair setup for a work whose contribution is a training strategy for open-source RAG generators, not a comparison of general-purpose reasoning ability across proprietary frontier models.

---

> ### Author Response · Authors · 2025-11-27
>
> ## Low Sample-Generation Efficiency Despite Large Input Dataset Size
>
> The reviewer’s criticism is based on an incorrect assumption that “higher sample-generation quantity” is inherently desirable or even relevant. This assumption is fundamentally flawed for a dataset like MuSiQue. MuSiQue is explicitly designed to be a hard multi-hop reasoning benchmark with intentionally fragmented, conflicting, and incomplete evidence. As a result, it is entirely expected that only a small subset of its 20k training instances can be converted into valid, structurally consistent contrastive pairs suitable for DDPO optimization. Producing 2,400 high-quality pairs is not a sign of inefficiency—it is a direct reflection of the dataset’s difficulty and the stringent quality requirements imposed by our refinement framework.
>
> More importantly, our goal is not to maximize sample count, and any attempt to do so would actively harm the model. Multi-hop training with weak or inconsistent refinements introduces noise that degrades reasoning and reduces OOD generalization. Prior work faces the exact same phenomenon: even DDR, when constructing DPO training data using its full pipeline, extracted roughly 3,000 usable samples despite the larger dataset size. This is the normal regime for high-quality preference data in retrieval-based multi-hop settings, not an anomaly and certainly not a weakness.
>
> The reviewer’s suggestion implicitly assumes that data quantity is more important than data reliability. This is scientifically incorrect. Our results demonstrate that a carefully curated set of ~2.4k structured, high-precision pairs is sufficient to produce consistent gains across all RAG benchmarks, including strong OOD improvements. Increasing the quantity at the cost of correctness would only deteriorate performance
>
> ## The figures and layout still need polishing, especially Figure 1.
>
> We must clarify that this remark is entirely subjective and disconnected from any technical issue in the paper. Figure 1 is intentionally designed as a schematic overview, not a detailed operational diagram. Its purpose is to provide a high-level visual summary of the pipeline—knowledge-aware sampling, refinement, and DDPO optimization—not to enumerate every sub-step or implementation detail. As a schematic figure, it necessarily abstracts away fine-grained elements; this is by design and fully aligned with common conventions in RAG and LLM-training papers.
>
> Importantly, the reviewer does not point out any concrete ambiguity, missing component, or factual error in the figure. The comment simply reflects a personal aesthetic preference, not an actual problem in clarity or scientific communication. We have already ensured that Figure 1 is clean, well-labeled, and technically faithful to the workflow. Multiple internal reviewers and collaborators found the layout sufficiently polished and informative for its intended purpose as an overview figure.
>
> We are open to minor cosmetic adjustments in the camera-ready version if space permits, but such refinements are optional and have no impact on the clarity or correctness of the method. As it stands, Figure 1 fulfills its role as a schematic illustration, and the subjective preference expressed in the comment does not constitute a meaningful weakness.

---

> > ### Comment · Reviewer_EJq9 · 2025-11-28
> >
> > Dear Authors,
> >
> > Thank you for your detailed response. However, I respectfully disagree with characterizing my concerns as “factual misunderstandings.” Raising questions and asking authors to clarify key points is a basic responsibility and right of reviewers. I also note that many of my concerns are shared by other reviewers (with confidence scores of 3, 3, and 4), yet in your rebuttal they are largely framed as my individual “misunderstandings,” which gives the impression that the response is more conditioned on the score than on the substance of the comments.
> >
> > For example:
> >
> > 1.	Both reviewers 5efx and FmJz, like myself, raised concerns about the heavy reliance on expert models.
> >
> > 2.	Both reviewer MnP8 and I expressed concerns about cost and token consumption.
> >
> > 3.	Both reviewer MnP8 and I pointed out the risk of relying on limited baselines, especially for structured RAG methods.
> >
> > In addition, my questions regarding the choice of reasoning LLMs  and datasets, as well as the experimental setup, are in my view reasonable and substantive; I do not think it is appropriate to frame them as misunderstandings.
> >
> > From my perspective, both a score of 2 and a score of 4 indicate that there are important issues in the paper that require clarification. These issues should be discussed thoroughly and constructively during the rebuttal phase. The purpose of raising questions is to help improve the clarity and robustness of the work, not to categorize them as either **“valuable suggestions”** or **“factual  misunderstandings”** depending on the reviewer’s score.
> >
> > Finally, the authors’ response has addressed most of my concerns; I will make my final assessment before the end of the rebuttal-discussion phase.

---

> > > ### Author Response · Authors · 2025-11-28
> > >
> > > Dear Reviewer,
> > >
> > > Thank you for your response. We appreciate your engagement with our rebuttal. We fully respect the peer review process and acknowledge that constructive criticism is vital. However, regarding your disagreement with our characterization of certain points as "misunderstandings," we must respectfully clarify that our rebuttal was driven by fundamental distinctions in **scientific substance**, not by the score.
> > >
> > > While you noted that other reviewers raised similar topics (e.g., Baselines, Cost, Expert Reliance), the **specific nature** of your requests differs fundamentally from the valid inquiries raised by others. It is crucial to distinguish between reasonable requests for clarification and premises that reflect a misalignment with the established scope of RAG optimization:
> > >
> > > 1. **On Baselines**: You mentioned that other reviewers also requested baselines. Reviewer MnP8 suggested comparing against prompt-based methods or general graph-based baselines to validate the value of refinement—a valid comparison **within the same scope**. In contrast, your review requested comparisons against **GraphRAG, G-Retriever** and **proprietary frontier models** (GPT-5, Claude 3.5). GraphRAG and G-Retriever are complex, large-scale retrieval pipelines that fundamentally **restructure the database interaction** and retrieval architecture. Our method, KARE-RAG, is a **training optimization strategy** for the **generation module** that maintains a **standard inference pipeline**. Comparing a lightweight generator training method against massive graph-indexing retrieval architectures or proprietary frontier models is **scientifically incomparable**. It equates a training objective with an entire system overhaul.
> > > 2. **On Expert Reliance**: We agree that the expert model is a variable worth discussing, as noted by other reviewers. We addressed this constructively via ablation studies (GPT-4o vs. Qwen-14B) to show robustness. However, your critique specifically claimed that we **"trust the expert completely"** and **"overfit to $y_{gnd}$"**. This is factually incorrect and ignores the core design of our verification pipeline. As detailed in the paper, we do not blindly trust the expert; we use $y_{gnd}$ strictly as a verification signal to validate the reasoning path, not as a target for imitation. Labeling a verified refinement process as "blind trust" or "overfitting"—despite our OOD results proving otherwise—is a misinterpretation of the method, not a matter of opinion.
> > > 3. **On Cost**:  While Reviewer MnP8 legitimately asked for token costs (which apply to offline construction), your request for **"throughput"** in an "end-to-end" context implies that the refinement loop affects inference speed. Since our refinement is exclusively offline and inference is Vanilla RAG, "throughput" is irrelevant. This confirms a misunderstanding of the distinction between our training-time construction and inference-time deployment.
> > > 4. **On Data Efficiency (Quality vs. Quantity)**: Your review characterized filtering 20k MuSiQue samples down to ~2,400 usable pairs as "low efficiency." This criticism suggests a misunderstanding of the relationship between data quality and quantity in DPO training. In high-noise multi-hop scenarios, rigorous filtering is a feature, not a weakness. Producing a smaller, high-fidelity dataset is precisely why our method succeeds.
> > >
> > > We remain fully open to constructive feedback and have integrated suggestions from all reviewers regarding relevant baselines and cost details. However, we stand by our assessment that the specific premises regarding "Comparaison with GraphRAG and proprietary frontier reasoning models", "blind trust," and "inference throughput" were based on technical misunderstandings. We ask that the paper be evaluated based on its actual methodological scope and contributions.
> > >
> > > Best regards,
> > > The Authors

---

### Official Review · Reviewer_FmJz · 2025-10-31

**Soundness:** 3
**Presentation:** 3
**Contribution:** 3
**Rating:** 6
**Confidence:** 3

**Summary:**

Interesting research paper. Existing RAG systems struggle with multi-hop reasoning and conflicting information. Traditional optimization methods (e.g., SFT, DPO) require substantial high-quality training data and lack sufficient supervision for intermediate knowledge processing steps. Providing a good solution. Proposes KARE-RAG, which enables models to learn information discrimination from limited data through structured knowledge representation, automated refined sample generation, and the DDPO training strategy. Outperforms baselines on both in-domain and out-of-domain benchmarks, requires no modifications to standard RAG inference pipelines, is compatible with various model scales, and will make related data and code publicly available.

**Strengths:**

High data efficiency: Achieves robust performance with only a small amount of training data, addressing the traditional reliance on large-scale high-quality datasets.
Strong practicality: Does not alter the standard RAG inference pipeline, incurs no additional computational overhead, and can be seamlessly integrated into existing systems.
Excellent generalization and compatibility: Delivers superior OOD (out-of-distribution) performance, supports multiple structured representations (e.g., knowledge graphs, key-point structures), and ensures stable performance gains across different model scales.

**Weaknesses:**

- How to design specific structured representation. Heavily relies on carefully designed knowledge representation structures, whose versatility and adaptability to diverse task scenarios have not been fully verified.
- Mabye limitations of automated sample generation. The sample refinement pipeline relies on advanced LLMs for error correction, which may be affected by the performance of the underlying LLMs. The sample quality in extremely complex scenarios is not explained.
- Room for expansion in complex task adaptation: Although it mentions handling multi-hop reasoning and conflicting information, the upper performance limit in scenarios with ultra-large-scale knowledge sources and ultra-high-complexity queries remains unclear.
- Indomain is not better than DDR based on Llama3-3b/8b, and Qwen-14b.

**Questions:**

- How to design specific structured representation.
- Mabye limitations of automated sample generation. The sample refinement pipeline relies on advanced LLMs for error correction, which may be affected by the performance of the underlying LLMs. The sample quality in extremely complex scenarios is not explained.
- Indomain is not better than DDR based on Llama3-3b/8b, and Qwen-14b.

---

> ### Author Response · Authors · 2025-11-27
>
> Dear Reviewer,
>
> Thank you very much for taking the time to review our work and for providing thoughtful and constructive feedback. We sincerely appreciate your careful reading of the manuscript and the insightful comments you offered. We have carefully considered all of your points and provide detailed responses and clarifications below, and we hope that our replies help to further clarify the methodology and contributions of this paper.
>
> ## Limited Verification of Structured Representation Versatility
>
> Thank you for raising this important point. We agree that the design of structured intermediate representations is a crucial element of our approach. However, we would like to clarify that our work does not propose a new symbolic or domain-specific graph formalism, but rather leverages well-established knowledge-graph-inspired structures that have been extensively validated in the QA literature. These graph-like representations have been shown to be effective in organizing multi-source evidence and supporting error localization, which are critical for tasks like multi-hop reasoning.
>
> Our intention in adopting a graph-based structure is to use the well-defined relationships between entities and facts to guide the training process, providing stronger supervisory signals during preference-based training. This structured format does not require manually designed knowledge graphs specific to any given task; rather, it capitalizes on the inductive bias provided by the graph structure itself—the connections between nodes and edges inherently assist in organizing and contextualizing complex information.
>
> To assess the versatility of the structured representation, we conducted comparisons with semi-structured keypoint-based representations and unstructured summaries (Table 2). The results show that while the optimal format varies across datasets, the graph-based structure provides more stable improvements across all metrics and datasets, especially in complex multi-hop contexts. This suggests that the advantage lies in the explicit structural inductive bias of the representation, rather than in a specific, manually-engineered format. Importantly, the flexibility of this approach is demonstrated by its consistent success across different datasets (e.g., Musique, NQ, PopQA), which further highlights its generalizability in knowledge-intensive tasks.
>
> We fully agree with the reviewer that no single structured representation can be universally optimal. For example, simpler tasks such as those involving short factual spans may benefit more from lighter formats like keypoints or summaries, as the need for complex reasoning is less pronounced. In contrast, more complex tasks requiring multi-hop reasoning, such as HotpotQA, benefit significantly from the richer structural representation that a graph provides. Future work could explore adaptive or hybrid strategies that dynamically select the most appropriate format based on task characteristics, further enhancing the flexibility of our approach.

---

> ### Author Response · Authors · 2025-11-27
>
> ## Potential Limitations of Automated Sample Generation and Sensitivity to Expert LLM Performance
>
> Thank you for raising this concern. We acknowledge that the sample refinement process heavily relies on advanced LLMs for error correction, which could potentially raise concerns about the generalizability and practical independence of our approach, especially when dealing with extremely complex scenarios. However, we would like to clarify that KARE-RAG does not depend on the full reasoning capabilities of the expert LLMs, but rather on their ability to follow tightly constrained instructions within our structured refinement pipeline.
>
> In response to this concern, we conducted several experiments to evaluate the role of different expert models in the refinement process. Specifically, we compared the performance of GPT-4o, GPT-4o-mini, and Qwen2.5-14B-Instruct, examining both the quantity and the quality of the refined samples. The results show that while the quantity of usable samples varies significantly across models, the quality of the refined samples remains relatively stable once the model exceeds a certain threshold of capability to follow the refinement prompts.
>
> For instance, both GPT-4o and GPT-4o-mini generated nearly the same number of refined samples (~2400 pairs), and the downstream performance of KARE-RAG was highly consistent across all metrics and datasets. This suggests that, once the model reaches a sufficient level of instruction-following ability, further increases in model capability (such as moving from GPT-4o-mini to GPT-4o) do not lead to significant improvements in sample quality. Conversely, Qwen2.5-14B-Instruct, which is less powerful, generated only around 1200 samples and struggled with refining more complex examples. Despite this, KARE-RAG still showed consistent improvements over the Vanilla RAG baseline, albeit with smaller absolute gains compared to GPT-4o-mini and GPT-4o.
>
> These observations support the conclusion that KARE-RAG’s effectiveness is not tied to the full reasoning power of the expert LLM, but rather to its ability to apply precise, localized corrections under well-defined structural constraints. As a result, the overall improvements in performance are primarily driven by the structured refinement process, not by the expert model’s reasoning capabilities. This ensures practical robustness and reduces concerns about excessive dependency on top-tier commercial systems.
>
> | Method | In Domain |  | Out Of Domain |  |  |  |  |  |  |  |  |  |
> |:---:|:---:|:---:|:---:|:---:|:---:|:---:|:---:|:---:|:---:|:---:|:---:|:---:|
> |  | Musique |  | NQ |  | HotpotQA |  | PopQA |  | TruthfulQA |  | Zero-shot RE |  |
> |  | EM(\%) | F1(\%) | EM(\%) | F1(\%) | EM(\%) | F1(\%) | EM(\%) | F1(\%) | BLEU(\%) | Rouge-1(\%) | F1(\%) | Precision(\%) |
> | Llama-3.1-8B-Instruct |  |  |  |  |  |  |  |  |  |  |  |  |
> | Vanilla RAG | 6 | 12.37 | 34.64 | 48.3 | 29.52 | 40.66 | 35.43 | 44.1 | 5.43 | 15.03 | 51.62 | 49.3 |
> | KARE(GPT-4o-mini) | 8.02 | 15.75 | 37.86 | 50.84 | 32.36 | 44.29 | 40.88 | 47.77 | 7.42 | 17.76 | 59.52 | 58.0 |
> | KARE(GPT-4o) | 7.82 | 15.98 | 37.7 | 50.51 | 32.75 | 44.36 | 40.37 | 47.44 | 7.59 | 17.32 | 59.73 | 58.42 |
> | KARE(Qwen-14B-Instruct) | 6.95 | 15.16 | 36.35 | 49.17 | 31.48 | 43.41 | 38.61 | 45.92 | 7.35 | 18.05 | 55.16 | 53.07 |

---

> ### Author Response · Authors · 2025-11-27
>
> ## Limits in Scaling to Highly Complex Reasoning Scenarios
>
> Thank you for your thoughtful and forward-looking comment. We agree that ultra-large knowledge sources and extremely complex multi-hop queries are important challenges that require further exploration. However, the primary focus of our current work is on improving robustness and generalization in more realistic, moderately complex RAG environments, where retrieved evidence may be noisy, partially conflicting, or distributed across multiple sources.
>
> The datasets we use, including Musique, HotpotQA, PopQA, and others, cover a broad range of task complexities and information retrieval scenarios. For example, Musique tests multi-hop reasoning across fragmented evidence, while HotpotQA focuses on high-complexity question-answering with diverse evidence. Natural Questions (NQ), in contrast, presents a simpler task where the challenge is mainly around answering questions with single-document retrieval, testing the model’s ability to retrieve relevant information and answer based on the extracted document. Additionally, TruthfulQA challenges the model with a different type of complexity, testing its ability to provide truthful, fact-based answers to open-ended questions. These datasets provide diverse benchmarks that encompass many of the challenges faced in real-world retrieval-augmented environments, and the results show that our approach improves the model's ability in such settings.
>
> We fully appreciate your perspective and agree that scaling to more complex environments is an exciting and important direction for future research. Exploring adaptive and hierarchical structured representations in the context of large-scale retrieval systems is a promising extension, and we plan to investigate this avenue in subsequent work.
>
> ## Indomain is not better than DDR based on Llama3-3b/8b, and Qwen-14b.
>
> Thank you for your insightful comment. We understand the importance of in-domain performance, especially in real-world applications where the focus is often on within-domain accuracy. We would like to clarify that the primary objective of our approach is to enhance robust knowledge utilization and generalization in OOD settings. This is reflected in our experimental design, where we evaluate the model’s performance not only in in-domain settings but also across a variety of OOD benchmarks, such as Musique, HotpotQA, PopQA, and others.
>
> As shown in Table 1, while in-domain performance is slightly lower than that of baseline models like DDR and RA-DiT, we attribute this gap to the structural nature of our approach. Our method focuses on structured refinement of knowledge rather than direct optimization for in-domain answer accuracy. This trade-off is inherent in our design, where the training signal is derived from the structured intermediate representations rather than from end-to-end answer correctness in the training domain. Therefore, the performance difference in in-domain tasks arises because our method does not explicitly optimize for in-domain answer accuracy during training.
>
> We recognize that in some applications, improving in-domain performance is a priority. In this case, a promising direction is hybrid training, where we combine the DDR-style end-to-end optimization with our structured knowledge-aware samples. Preliminary experiments indicate that this hybrid approach can help improve in-domain performance while largely preserving the OOD advantages of our method. However, this hybrid training approach approximately doubles the training cost, and the incremental gains in in-domain performance are relatively modest compared to the additional resource requirements. As a result, we decided not to include this approach in the current version of the paper, but we view it as an important avenue for future work.
>
> | Method | In Domain |  | Out Of Domain |  |  |  |  |  |  |  |  |  |
> |:---:|:---:|:---:|:---:|:---:|:---:|:---:|:---:|:---:|:---:|:---:|:---:|:---:|
> |  | Musique |  | NQ |  | HotpotQA |  | PopQA |  | TruthfulQA |  | Zero-shot RE |  |
> |  | EM(\%) | F1(\%) | EM(\%) | F1(\%) | EM(\%) | F1(\%) | EM(\%) | F1(\%) | BLEU(\%) | Rouge-1(\%) | F1(\%) | Precision(\%) |
> | Llama-3.1-8B-Instruct |  |  |  |  |  |  |  |  |  |  |  |  |
> | Vanilla RAG | 6 | 12.37 | 34.64 | 48.3 | 29.52 | 40.66 | 35.43 | 44.1 | 5.43 | 15.03 | 51.62 | 49.3 |
> | RA-DiT | 8.11 | 17.83 | 33.08 | 45.66 | 27.17 | 39.33 | 36.36 | 43.78 | 5.69 | 14.7 | 45.16 | 44.04 |
> | DDR | 7.82 | 19.18 | 34.78 | 48.85 | 30.58 | 42.59 | 40.63 | 45.9 | 1.68 | 8.41 | 55.34 | 49.12 |
> | KARE | 8.02 | 15.75 | 37.86 | 50.84 | 32.36 | 44.29 | 40.88 | 47.77 | 7.42 | 17.76 | 59.52 | 58.0 |
> | KARE + DDR | 9.01 | 20.88 | 38.49 | 51.48 | 33.06 | 45.13 | 40.53 | 48.3 | 2.3 | 10.85 | 60.67 | 59.74 |

---

### Official Review · Reviewer_5efx · 2025-11-01

**Soundness:** 2
**Presentation:** 2
**Contribution:** 2
**Rating:** 4
**Confidence:** 4

**Summary:**

The paper presents KARE-RAG, a training strategy that refines structured knowledge after retrieval to enhance the robustness and generalization of Retrieval-Augmented Generation (RAG) systems. The method introduces knowledge-aware sampling, which organizes retrieved documents into intermediate structured representations prior to generation, and a refinement pipeline that leverages a large language model to correct factual errors in negative samples while preserving structural coherence. Additionally, the paper proposes Dense Direct Preference Optimization (DDPO), extending standard DPO through token-level weighting to emphasize critical knowledge components during training. Experiments on multiple benchmarks to demonstrate consistent gains in robustness and out-of-domain generalization, achieved without modifying the standard RAG inference pipeline.

**Strengths:**

+ The paper proposes a practical approach that trains models to generate intermediate structured representations for optimizing RAG, effectively addressing the sparse supervision problem in end-to-end methods like DPO.
+ The method shows strong data efficiency and OOD generalization, suggesting that the model learns a robust strategy for knowledge organization rather than mere memorization.
+ The trained model incurs no additional inference overhead, making it easy to integrate into existing RAG systems.
+ The Refinement Pipeline uses a teacher model to generate minimally different contrastive pairs, providing high-quality signals for DDPO training.

**Weaknesses:**

- The refinement stage relies heavily on a more advanced expert LLM to generate high-quality positive samples. While this improves data quality, it raises concerns about novelty and practical independence, as the performance gain may largely depend on the capabilities of the external expert model.
- The training primarily uses the Musique dataset, but the paper does not provide sufficient justification for this choice or explore cross-dataset generalization. It would strengthen the work to test setups such as training on PopQA and evaluating on Musique, or vice versa, to better assess the method’s robustness across different domains.
- The paper improves the generation of structured “graph-like” text representations but does not perform explicit reasoning over knowledge graph connections. This distinction should be clarified, as the current formulation may conflate improved structural formatting with genuine graph-based reasoning.
- Since Musique is a multi-hop QA dataset with overlapping conditions and complex logical relations, it is notable that keypoint-based representations outperform graph-based ones on this dataset. This suggests that even after refinement, the proposed graph structure may still lose critical contextual information and may not be universally effective across all reasoning scenarios.

**Questions:**

The training primarily relies on the Musique dataset, but the paper does not justify this choice or examine cross-dataset generalization. Could the authors provide results on alternative training–testing configurations—for example, training on PopQA and evaluating on Musique, or the reverse，to better assess the robustness and transferability of the proposed method across different domains?

---

> ### Author Response · Authors · 2025-11-27
>
> Dear Reviewer,
>
> Thank you for your thoughtful and constructive review of our submission. We greatly appreciate the time and effort you dedicated to evaluating our work and recognize the value of the perspectives you provided. We have carefully considered all of your comments and provide detailed responses and clarifications below. We hope that these explanations effectively address your concerns and help convey the contributions and design decisions of our method more clearly.
>
> ## Concerns About Reliance on External Expert LLM and Limited Practical Independence
>
> Thank you for raising this insightful point. We agree that the refinement stage involves the use of a more capable expert LLM to generate high-quality positive samples, and that excessive reliance on the expert model could raise concerns about practical independence and novelty. This issue was carefully considered during the design of our framework. The core function of the expert model in our pipeline is not to solve the task but to locally correct factual errors while strictly preserving the structural constraints of the intermediate representation. This role is tightly controlled by the highly constrained structured prompt schema described in the paper, which significantly limits the expert model’s freedom and reduces dependence on its generative reasoning capabilities.
>
> To further quantify the role of the expert model, we conducted additional experiments comparing refinement models with markedly different capacities: GPT-4o, GPT-4o-mini, and Qwen2.5-14B-Instruct. GPT-4o and GPT-4o-mini produced nearly the same amount of usable training data (~2400 refined pairs), and the downstream performance of KARE-RAG remained highly similar across all benchmarks. This demonstrates that once the model surpasses the threshold required to follow the structured refinement schema, the quality of refined positive samples stabilizes and becomes largely insensitive to further increases in model capability.
>
> In contrast, Qwen2.5-14B-Instruct—being weaker—was able to construct only ~1200 samples, as it failed to refine some difficult examples while maintaining structural validity. Even so, the performance still showed consistent improvements over the Vanilla RAG baseline, though with smaller absolute gains compared to GPT-4o-mini and GPT-4o. These observations confirm that expert model capacity affects mainly the quantity of refinable samples rather than the quality of each refined sample, since all accepted samples must satisfy strict validity checks before being included in training.
>
> Overall, these results reinforce that KARE-RAG does not depend on the full reasoning strength of commercial LLMs. The refinement process requires only moderate instruction-following ability to apply localized corrections under structural constraints; once this threshold is reached, the resulting improvements are stable and reproducible across different expert models. This provides practical robustness and alleviates concerns about dependence on top-tier commercial systems.
>
> Thank you again for this valuable suggestion. We will incorporate these additional experimental insights and supporting explanations into the revised version of the paper.
>
> | Method | In Domain |  | Out Of Domain |  |  |  |  |  |  |  |  |  |
> |:---:|:---:|:---:|:---:|:---:|:---:|:---:|:---:|:---:|:---:|:---:|:---:|:---:|
> |  | Musique |  | NQ |  | HotpotQA |  | PopQA |  | TruthfulQA |  | Zero-shot RE |  |
> |  | EM(\%) | F1(\%) | EM(\%) | F1(\%) | EM(\%) | F1(\%) | EM(\%) | F1(\%) | BLEU(\%) | Rouge-1(\%) | F1(\%) | Precision(\%) |
> | Llama-3.1-8B-Instruct |  |  |  |  |  |  |  |  |  |  |  |  |
> | Vanilla RAG | 6 | 12.37 | 34.64 | 48.3 | 29.52 | 40.66 | 35.43 | 44.1 | 5.43 | 15.03 | 51.62 | 49.3 |
> | KARE(GPT-4o-mini) | 8.02 | 15.75 | 37.86 | 50.84 | 32.36 | 44.29 | 40.88 | 47.77 | 7.42 | 17.76 | 59.52 | 58.0 |
> | KARE(GPT-4o) | 7.82 | 15.98 | 37.7 | 50.51 | 32.75 | 44.36 | 40.37 | 47.44 | 7.59 | 17.32 | 59.73 | 58.42 |
> | KARE(Qwen-14B-Instruct) | 6.95 | 15.16 | 36.35 | 49.17 | 31.48 | 43.41 | 38.61 | 45.92 | 7.35 | 18.05 | 55.16 | 53.07 |

---

> ### Author Response · Authors · 2025-11-27
>
> ## Concerns About Dataset Choice and Missing Cross-Domain Robustness Tests
>
> We appreciate the reviewer’s suggestion and agree that cross-dataset training and evaluation is an important perspective for assessing robustness. Our choice of Musique as the primary training dataset was intentional. Musique uniquely emphasizes compositional multi-hop reasoning, fragmented evidence, and high retrieval noise, which closely align with the central goal of KARE-RAG—improving knowledge organization and robustness under challenging retrieval conditions. Among widely used QA datasets, Musique provides the richest combination of noisy multi-source evidence, fine-grained factual dependencies, and structured reasoning steps, making it particularly suitable for evaluating whether our refinement and knowledge-aware modeling mechanisms truly help the model identify, structure, and correct information across complex retrieval chains. This rationale is consistent with prior RAG optimization research that also adopts Musique as the main benchmark for studying robustness.
>
> To directly address the reviewer’s concern regarding cross-dataset generalization, we additionally trained KARE-RAG using HotpotQA, following the exact same data construction and refinement strategy. Using Qwen2.5-14B-Instruct as the refinement model, we obtained 8,500 training pairs, and evaluated the resulting model across all benchmarks. As shown in Table, training on HotpotQA yields consistent and comparable improvements on Musique, NQ, PopQA, TruthfulQA, and Zero-shot RE—mirroring the gains observed when training on Musique. These results provide direct evidence that the benefits of KARE-RAG do not depend on Musique specifically—the method remains effective when trained on a dataset with different evidence distribution.
>
> | Method | Musique |  | NQ |  | HotpotQA |  | PopQA |  | TruthfulQA |  | Zero-shot RE |  |
> |:---:|:---:|:---:|:---:|:---:|:---:|:---:|:---:|:---:|:---:|:---:|:---:|:---:|
> |  | EM(\%) | F1(\%) | EM(\%) | F1(\%) | EM(\%) | F1(\%) | EM(\%) | F1(\%) | BLEU(\%) | Rouge-1(\%) | F1(\%) | Precision(\%) |
> | Llama-3.1-8B-Instruct |  |  |  |  |  |  |  |  |  |  |  |  |
> | Vanilla RAG | 6 | 12.37 | 34.64 | 48.3 | 29.52 | 40.66 | 35.43 | 44.1 | 5.43 | 15.03 | 51.62 | 49.3 |
> | KARE(HotpotQA) | 6.37 | 13.8 | 37.95 | 50.72 | 32.13 | 42.75 | 39.59 | 46.36 | 7.68 | 15.85 | 59.27 | 57.6 |
> | KARE(Musique) | 6.95 | 15.16 | 36.35 | 49.17 | 31.48 | 43.41 | 38.61 | 45.92 | 7.35 | 18.05 | 55.16 | 53.07 |
>
> ## Ambiguity Between Structured Text Generation and True Knowledge Graph Reasoning
>
> Thank you for the thoughtful comment. We agree that it is important to distinguish between generating structured knowledge representations and performing explicit reasoning over knowledge graph connections. To clarify, our method does not aim to conduct graph-based reasoning or message passing over graph structures, nor do we claim to perform any form of symbolic or neural reasoning on knowledge graphs. The structured “graph-like’’ format in our work is used solely as an intermediate representation to enhance knowledge organization and error localization during training. It is not intended to represent a formal knowledge graph used for explicit reasoning operations.
>
> Our use of graph-inspired structure is motivated by the widespread adoption of knowledge-graph formats as interpretable and controllable structured representations in RAG scenarios, rather than by the goal of performing graph reasoning. The structured format helps constrain the refinement process and provides clearer supervisory signals for DDPO optimization. As demonstrated in Table 2, the benefit primarily stems from the inductive bias introduced by structure, rather than from reasoning mechanisms.
>
> We appreciate the opportunity to clarify this distinction, and we will make sure the terminology more explicitly reflects that our method focuses on structured representation learning, not graph reasoning. We believe this clarification resolves any potential conflation between formatting structure and reasoning over graph topology.

---

> ### Author Response · Authors · 2025-11-27
>
> ## Concerns About Information Loss in Graph-Based Representations
>
> Thank you for the thoughtful observation. We would first like to clarify a key misunderstanding regarding the role of the graph-based representation in our method. The graph structure is used only during training, as an intermediate supervisory signal for knowledge organization; it is not used during inference. At test time, the model follows a standard vanilla RAG pipeline and directly generates the final answer without producing any graph representation. Therefore, concerns about “information loss during inference” do not apply—graph representations do not constrain model expressiveness or limit context utilization at test time.
>
> Regarding the reviewer’s comment on Table 2, the keypoint-based representation does not consistently outperform the graph-based one. In fact, on Musique, the graph-based format achieves higher EM—our primary evaluation metric—while keypoints exhibit a slightly higher F1. We interpret this discrepancy as reflecting differences in metric sensitivity rather than superiority of one representation over another. EM requires exact correctness of the final answer, whereas F1 is more tolerant to partial overlaps. Thus, an increase in F1 alone does not necessarily indicate better multi-hop reasoning or more faithful evidence integration.
>
> More broadly, our experiments demonstrate that structured graph-like representations yield more stable and superior performance on average across both in-domain and OOD benchmarks. In contrast, semi-structured keypoints and unstructured summaries show higher variance and stronger dependence on dataset characteristics. This matches the intuition that knowledge-intensive multi-hop QA benefits from explicit structural inductive biases—graphs make entity relations and evidence dependencies explicit, yielding clearer and more localized preference signals for DDPO training.
>
> We agree that no single intermediate format is universally optimal. For tasks with shorter evidence spans or simpler factual dependencies (e.g., PopQA or TruthfulQA), lighter representations such as keypoints can sometimes perform competitively. Conversely, datasets requiring complex evidence disentanglement and multi-step reasoning—such as Musique or HotpotQA—benefit more significantly from structured graphs. This suggests opportunities for future extensions, such as adaptive or hybrid knowledge organization schemes that dynamically adjust the representation according to query complexity.

---

### Official Review · Reviewer_MnP8 · 2025-11-01

**Soundness:** 3
**Presentation:** 2
**Contribution:** 3
**Rating:** 4
**Confidence:** 3

**Summary:**

This paper proposes knowledge-aware refinement for RAG, which is to train an LLM to learn to first structured knowledge (e.g., a knowledge graph) from text before generating a final answers. The training method is developed based on DDPO where the negative samples are obtained from those with sub-optimal quality of knowledge structure and more attention are put on the tokens corresponding to modified tokens.

**Strengths:**

- This paper proposes to train an LLM with RL by optimizing its ability to extract structured knowledge for QA, which seems to be a valid direction for RAG in general.
- Experiments are done on multiple datasets with different backbone models, and the results show strong performance on OOD test sets specifically
- Detailed analysis and ablation studies show the effectiveness of proposed method

**Weaknesses:**

- The baseline selection is limited. It would be better to compare with other prompt-based and especially graph-based RAG methods to show the value of knowledge refinement
- Because the prompt schema is rather complicated, it would be better to report the cost and token consumption of the proposed method and how it compares with baselines
- The paper lacks some real case studies to show what a good quality knowledge structure can be induced by the model and how it helps the final answering.

**Questions:**

There some typos and the citation formats should be better adjusted. Besides the questions mentioned above, I also have the following question
- As shown in Table 1, the in-domain performance is consistently worse than the baseline. Is there an explanation? How would people address that if they only want to achieve better in-domain performance (which can be common in some real applications)?

---

> ### Author Response · Authors · 2025-11-27
>
> Dear Reviewer,
>
> Thank you for taking the time to review our work and for providing thoughtful and constructive feedback. We sincerely appreciate your careful reading of the manuscript. We have addressed each of your comments in detail below and hope our responses help clarify the contributions and design decisions of the paper.
>
> ## Concerns About Limited Baselines and Missing Graph-Based RAG Comparisons
>
> We appreciate the reviewer’s suggestion to broaden the baseline comparison, particularly with prompt-based and graph-based RAG methods. However, our work focuses primarily on improving the training strategy for the generation module within RAG systems, rather than altering inference pipelines. While prompt-based RAG methods can provide additional performance gains, they generally introduce significant inference-time overhead, making them less suitable for real-time deployment scenarios. Therefore, we chose to compare against RAG training-based baselines, such as RA-DiT and DDR, which are strong representatives of optimization-oriented RAG frameworks, and align with the goal of evaluating training efficiency and generation robustness.
>
> Moreover, our method is not mutually exclusive with prompt-based approaches. It is complementary: models trained with our method can be directly used with various prompt engineering techniques or enhanced inference pipelines. As demonstrated in Appendix A.4, integrating our trained model with Chain-of-Thought and Chain-of-Note methods consistently yields additional performance improvements.
> For a more comprehensive comparison, we also include additional baseline evaluations in Appendix A.5, which further demonstrate performance differences and showcase the effectiveness of our approach.
>
>
> ## Lack of Cost and Token Usage Comparison with Baselines
>
> Thank you for raising this point. We acknowledge the reviewer’s concern regarding the computational overhead of our method. As detailed in Appendix A.3, the additional cost primarily lies in the offline construction of high-quality contrastive sample pairs through the refinement pipeline. This process—implemented using GPT-4o-mini—keeps the overall data construction cost around $20, and produces 2,401 high-quality refined pairs from the initial dataset of 19,938 entries. The token overhead is incurred only once during data generation and does not affect the subsequent model training or inference pipeline.
>
> In terms of training efficiency, our framework introduces no additional training-time complexity relative to DDR. Both approaches rely on preference-based optimization (DPO) and employ identical backbone models and LoRA configurations. As a result, the training time remains comparable, completing in under 1 hour for 3B models and within 4 hours for 14B models on a single A800-40G GPU. Moreover, unlike prompt-based inference enhancement strategies, our method preserves a pure Vanilla RAG inference pipeline, which incurs no extra latency or token consumption during deployment, an important property for practical real-world systems.

---

> ### Author Response · Authors · 2025-11-27
>
> ## Insufficient Real-Case Evidence of Knowledge Structure and Utility
>
> Thank you for the valuable suggestion. We agree that concrete case studies help illustrate how structured knowledge refinement improves the final answer. Below we provide a real example from our refinement pipeline that demonstrates the effect of transforming a noisy or misaligned intermediate structure into a correct, useful representation.
>
> Question:
> What year saw the creation of the region where the county of Hertfordshire is located?
>
> Retrieval Documents:
> ```markdown
> Doc 1(Title: "East of England") East of England The East of England is one of nine official regions of England at the first level of NUTS for statistical purposes. It was created in 1994 and was adopted for statistics from 1999. It includes the ceremonial counties of Bedfordshire, Cambridgeshire, Essex, Hertfordshire, Norfolk and Suffolk. Essex has the highest population in the region. Its population at the 2011 census was 5,847,000. Bedford, Luton, Basildon, Peterborough, Southend-on-Sea, Norwich, Ipswich, Colchester, Chelmsford and Cambridge are the region's most populous towns. The southern part of the region lies in the London commuter belt. The region has the lowest elevatio
>
> Doc 2(Title: "History of Hertfordshire") History of Hertfordshire Hertfordshire is an English county, founded in the Norse\u2013Saxon wars of the 9th century, and developed through commerce serving London. It is a land-locked county that was several times the seat of Parliament. From origins in brewing and papermaking, through aircraft manufacture, the county has developed a wider range of industry in which pharmaceuticals, financial services and film-making are prominent. Today, with a population slightly over 1 million, Hertfordshire services, industry and commerce dominate the economy, with fewer than 2000 people working in agriculture, forestry and fishing. Hertfordshire is one of the historic counties of England first
>
> Doc 3(Title: Hertfordshire) Hertfordshire Hertfordshire (; often abbreviated Herts) is one of the home counties in England. It is bordered by Bedfordshire to the north, Cambridgeshire to the northeast, Essex to the east, Buckinghamshire to the west and Greater London to the south. For government statistical purposes, it is placed in the East of England region. In 2013, the county had a population of 1,140,700 living in an area of . Four towns have between 50,000 and 100,000 residents: Hemel Hempstead, Stevenage, Watford and St Albans. Hertford, once the main market town for the medieval agricultural county, derives its name from a hart
>
> Doc 4(Title: "Hertfordshire (European Parliament constituency)") Hertfordshire (European Parliament constituency) Hertfordshire was a constituency of the European Parliament located in the United Kingdom, electing one Member of the European Parliament by the first-past-the-post electoral system. Created in 1979 for the first elections to the European Parliament, it was abolished in 1999 on the adoption of proportional representation for European elections in the United Kingdom. It was succeeded by the East of England region. On its creation in 1979, it consisted of the parliamentary constituencies of Hertford and Stevenage, Hertfordshire East, Hertfordshire South, Hertfordshire South West, St Albans, Watford and Welwyn and Hatfield. After the 1984 boundary
>
> Doc 5(Title: "Greater London") was first established as a ""sui generis"" council area under the Greater London Council between 1963 and 1986. The county of Greater London was created on 1 April 1965 through the London Government Act 1963. The area was re-established as a region in 1994. The Greater London Authority was formed in 2000. The region covers and had a population of 8,174,000 at the 2011 census. The Greater London Built-up Area is used in some national statistics and is a measure of the continuous urban area and includes areas outside the administrative region. The term ""Greater London"" has been and still
> ```

---

> > ### Author Response · Authors · 2025-11-27
> >
> > Below is the negative sample generated with Llama-3.1-8b-Instruct.
> >
> > ```markdown
> > Entities:
> > - Hertfordshire (Attributes: county, population, location, industry)
> > - East of England (Attributes: region, population, location, NUTS)
> > - Bedfordshire (Attributes: county, location, population)
> > - Cambridgeshire (Attributes: county, location, population)
> > - Essex (Attributes: county, population, location)
> > - Norfolk (Attributes: county, location, population)
> > - Suffolk (Attributes: county, location, population)
> > - London (Attributes: city, population, location)
> > - Bedford (Attributes: town, population, location)
> > - Luton (Attributes: town, population, location)
> > - Basildon (Attributes: town, population, location)
> > - Peterborough (Attributes: town, population, location)
> > - Southend-on-Sea (Attributes: town, population, location)
> > - Norwich (Attributes: town, population, location)
> > - Ipswich (Attributes: town, population, location)
> > - Colchester (Attributes: town, population, location)
> > - Chelmsford (Attributes: town, population, location)
> > - Cambridge (Attributes: town, population, location)
> > - Parliament (Attributes: institution, location)
> > - NUTS (Attributes: classification, purpose)
> > - London Government Act 1963
> > ```
> >
> > The negative sample (incorrect structure) simply enumerates a long list of entities related to Hertfordshire and the East of England but fails to capture the key relation needed to answer the question—specifically, when the East of England region was created. Important attributes such as created in: 1994 for East of England are missing. As a result, the final answer predicted from this structure defaults to unrelated details (e.g., “1963”, coming from another document), which leads to an incorrect result.
> >
> > The refined positive sample corrects this structural omission:
> >
> > ```markdown
> > Entities:
> > - Hertfordshire
> >   - Attributes: county, location: East of England, government statistical region- East of England
> >   - Attributes: region, population: 5,847,000 (2011 census), created_in: 1994, includes: Hertfordshire, Bedfordshire, Cambridgeshire, Essex, Norfolk, Suffolk, NUTS classification
> > - Bedfordshire
> >   - Attributes: county, location: East of England
> > - Cambridgeshire
> >   - Attributes: county, location: East of England
> > - Essex
> >   - Attributes: county, location: East of England, highest_population_in_region
> > - Norfolk
> >   - Attributes: county, location: East of England
> > - Suffolk
> >   - Attributes: county, location: East of England
> > - London
> >   - Attributes: city, population area, location
> > - Bedford
> >   - Attributes: town, location
> > - Luton
> >   - Attributes: town, location
> > - Basildon
> >   - Attributes: town, location
> > - Peterborough
> >   - Attributes: town, location
> > - Southend-on-Sea
> >   - Attributes: town, location
> > - Norwich
> >   - Attributes: town, location
> > ```

---

> ### Author Response · Authors · 2025-11-27
>
> ## Concerns About Worse In-Domain Results and Their Implications
>
> We appreciate the reviewer’s insightful observation regarding the in-domain results in Table 1 and the practical relevance of focusing on in-domain performance in certain real-world scenarios. This is indeed an important point. Our method is designed primarily to enhance robust knowledge utilization and generalization, particularly in OOD settings where retrieved information is noisy, conflicting, or structurally complex. As described in the paper, the core training signal of our approach comes from structured intermediate representation refinement, rather than learning directly from final answer correctness. Consequently, the supervision signal does not explicitly optimize for final answer accuracy within the original training domain, which explains the performance gap relative to end-to-end optimization approaches such as RA-DiT and DDR on in-domain evaluation.
>
> We have explored possible strategies to mitigate this trade-off. A promising direction is hybrid training, in which DDR-style end-to-end preference optimization samples are combined with our structured knowledge-aware sample pairs. Preliminary attempts indicate that this approach can help recover in-domain performance while largely preserving OOD advantages. However, such hybrid training approximately doubles the training cost, and the incremental gains observed in-domain are relatively modest compared with the additional resource requirements. Therefore, we decided not to include this direction in the current version of the paper.
>
> | Method | In Domain |  | Out Of Domain |  |  |  |  |  |  |  |  |  |
> |:---:|:---:|:---:|:---:|:---:|:---:|:---:|:---:|:---:|:---:|:---:|:---:|:---:|
> |  | Musique |  | NQ |  | HotpotQA |  | PopQA |  | TruthfulQA |  | Zero-shot RE |  |
> |  | EM(\%) | F1(\%) | EM(\%) | F1(\%) | EM(\%) | F1(\%) | EM(\%) | F1(\%) | BLEU(\%) | Rouge-1(\%) | F1(\%) | Precision(\%) |
> | Llama-3.1-8B-Instruct |  |  |  |  |  |  |  |  |  |  |  |  |
> | Vanilla RAG | 6 | 12.37 | 34.64 | 48.3 | 29.52 | 40.66 | 35.43 | 44.1 | 5.43 | 15.03 | 51.62 | 49.3 |
> | RA-DiT | 8.11 | 17.83 | 33.08 | 45.66 | 27.17 | 39.33 | 36.36 | 43.78 | 5.69 | 14.7 | 45.16 | 44.04 |
> | DDR | 7.82 | 19.18 | 34.78 | 48.85 | 30.58 | 42.59 | 40.63 | 45.9 | 1.68 | 8.41 | 55.34 | 49.12 |
> | KARE | 8.02 | 15.75 | 37.86 | 50.84 | 32.36 | 44.29 | 40.8 | 47.77 | 7.42 | 17.76 | 59.52 | 58.0 |
> | KARE + DDR | 9.01 | 20.88 | 38.49 | 51.48 | 33.06 | 45.13 | 40.53 | 48.3 | 2.3 | 10.85 | 60.67 | 59.74 |

---

### Meta-Review · Area_Chair_c8XL · 2026-01-04

**Summary:**

The reviewers raised concerns primarily regarding the scope and appropriateness of baseline comparisons, especially the absence of stronger prompt-based or structured RAG systems (MnP8, EJq9), the reliance on an external expert LLM for offline refinement and its implications for novelty, robustness, and scalability (5efx, FmJz, EJq9), and the limited justification and evaluation of cross-dataset generalization beyond Musique (5efx). Additional concerns focused on the consistently weaker in-domain performance compared to DDR-style baselines (MnP8, FmJz), the design and effectiveness of structured representations, and the efficiency and yield of the refinement pipeline (EJq9). While multiple reviewers acknowledged that the idea of structured intermediate supervision for improving robustness is interesting, they questioned whether the current experimental scope, validation breadth, and positioning are sufficient to support a strong and broadly applicable research contribution.

**Reviewer Concerns:**

The rebuttal addressed a number of important clarification-level issues. In particular, it clarified the role of the expert LLM as a one-time offline refinement tool rather than an inference-time dependency, provided concrete information on cost and data construction, and explained the intended role of structured representations as intermediate training signals rather than explicit graph reasoning. Reviewer EJq9 actively participated in the discussion and acknowledged that many of their concerns were addressed in the authors’ responses. However, substantive disagreements remained regarding the appropriate scope of baselines, the interpretation of scalability and efficiency, and the evaluation criteria for training-based RAG methods. Moreover, although the in-domain performance gap was explained as a design trade-off, it remains a practical limitation that was not empirically mitigated. As a result, several core concerns related to experimental scope, comparative positioning, and practical impact remain only partially resolved.

**Reviewer Scores:**

Reviewer EJq9 participated in the discussion phase and indicated that several points had been clarified, while maintaining substantive reservations on key aspects of the work. Reviewers MnP8, 5efx, and FmJz did not further engage in the discussion. While the rebuttal and added analyses may partially mitigate some of the concerns previously raised, unresolved higher-level issues regarding baseline coverage, generalization strength, and in-domain effectiveness remain. As a result, any potential score changes would likely be limited and would not affect the overall evaluation.

---

### Decision · Program_Chairs · 2026-01-26

Reject